# Technological, Microstructural and Strength Aspects of Welding and Post-Weld Heat Treatment of Martensitic, Wear-Resistant Hardox 600 Steel

**DOI:** 10.3390/ma14164541

**Published:** 2021-08-12

**Authors:** Łukasz Konat

**Affiliations:** Department of Vehicle Engineering, Wroclaw University of Science and Technology, 50-370 Wroclaw, Poland; lukasz.konat@pwr.edu.pl

**Keywords:** wear-resistant martensitic steel, welding design, submerged arc welding (SAW), post-weld heat treatment, structures, hardness changes, mechanical properties, Hardox 600 steel, fractographic analysis

## Abstract

The study presents technological and structural aspects of production and heat treatment of welded joints of high-strength, abrasion-resistant Hardox 600 steel. As a result of the conducted research, it was found that the use of welding processes for joining this steel leads to the formation of a wide heat-affected zone, characterized by various structures favoring the reduction of abrasion resistance and deterioration of plastic properties, while increasing the susceptibility to brittle fracture. On the basis of the structural and strength characteristics, an effective welding technology for Hardox 600 steel was proposed, as well as the conditions and parameters of post-welding heat treatment, leading to obtaining structures close to a base material in the entire area of the welded joint. Despite the limited high-carbon equivalent CEV of the metallurgical weldability of the tested steel, the tests carried out in laboratory conditions allowed researchers to obtain welded joints characterized by very high strength indexes, corresponding to the base material, while maintaining satisfactory plastic and impact properties.

## 1. Introduction

Hardox 600 steel belongs to the group of alloys defined by their producer, i.e., the Swedish steelworks SSAB, as weldable, martensitic steel resistant to abrasive wear with boron [1]. A very small number of available scientific publications on this steel, especially in the context of its weldability, prompted the author’s interest in this material group. It is worth noting that Hardox 600 steel, apart from Hardox Extreme, stands out significantly from other steels offered by the Swedish steel company, as well as by other steel producers with properties similar to Hardox steel. It is characterized by an average hardness—depending on the thickness of the sheet—exceeding even 600 units on the Brinell scale and a very high value of static strength—R_m_ ≥ 2000 MPa [2]. Therefore, there are reasonable premises proving the possibility of using the high strength indexes of Hardox 600 steel for selected devices and machine parts, structural elements of which are joined by welding technologies. Very often, in this type of application, in addition to high tensile strength, engineering plastics must have an appropriate level of resistance to impact loads. The Charpy test, most often used to describe this parameter, is not always able to truly reflect the material’s resistance to variable dynamic loads. The impact toughness value is only a comparative indicator for materials with similar properties simultaneously tested under the same conditions. Therefore, in order to obtain more objective research results, fractographic analyzes were also used. Welded joints of Hardox 600 steel in the delivered condition (directly after welding) and after the performed post-welding heat treatment were tested. The aim of these actions was an attempt to recreate in laboratory conditions structural and mechanical properties of the entire welded joint, corresponding to the base material. It is also worth mentioning that, in relation to Hardox 600 steel, such attempts have already been made by the authors of [3]. Nevertheless, they were based on the use of manual GTAW welding, characterized by a low degree of repeatability, and in addition, according to the author, their results were not fully satisfactory. Therefore, in this study, a broader description of the functional and manufacturing properties of Hardox 600 steel, which was already carried out in the above-mentioned study and in the author’s own study [4], has been abandoned, and attention has been focused mainly on the presentation of the results of this steel, focused on the use of welding technologies. Detailed chemical compositions and strength properties of Hardox 600 are presented in Table 1, Table 2 and Table 3.

Considering the possibility of using welding techniques to join selected Hardox steels, it can be concluded that numerous of the author’s own reconnaissance studies, e.g., [6,7,8,9,10,11], as well as literature data [12], indicate good weldability of Hardox 400 and 450 steels and moderately satisfactory weldability of Hardox 500. In each of the cited cases, the obtained welded joints did not show any significant macroscopic and microscopic welding imperfections, while at the same time being characterized—in relation to popular, low-carbon, low-alloy structural steels—with relatively high strength indexes. However, regardless of the obtained properties, in all considered welded joints, in the heat-affected zone, unfavorable changes in the structure were observed, resulting in a reduction in resistance to abrasive wear—a basic functional feature of the discussed material group. In order to eliminate this phenomenon, in [13], it is proposed to apply additional heat treatment after the welding process, in relation to low-alloy abrasion-resistant steels with a martensitic structure. These measures would be aimed at eliminating the discussed structural changes, causing unfavorable hardness courses in the entire area of the welded joint and reducing the resistance to abrasive wear processes. It is also worth noting that the use of additional post-welding heat treatments significantly extends the possibilities of using welding techniques in relation to this material group, with particular emphasis on steels with increased carbon content. The development of the above-mentioned issues seems to be also justified in the context of large discrepancies in the weldability of these steels, which occur between the data published in the material cards and the results of the author’s own research. Most often, this problem results from imprecise chemical composition, and thus the inability to determine the actual equivalent of metallurgical weldability (CEV) or carbon equivalent (CET), most often determining the parameters of pre-heating steel before welding. It is true that the producers of these steels most often provide information on the typical and maximum value of both carbon equivalents, related to a given range of available sheet thicknesses; nevertheless, the results of numerous of the author’s own tests (cited above), as well as the results included in Table 2, clearly indicate the lack of correlation of these data with the actual chemical compositions and the CEV and CET equivalents calculated on their basis. Therefore, numerous attempts made by the exploiters of low-alloy steels resistant to abrasive wear to use their very high strength indices in welded structures have not produced satisfactory results. The result of this state of affairs is complete abandonment of the use of steel grades of this type, characterized by the highest strength parameters, in favor of their weaker counterparts, characterized by experimentally proven good or satisfactory weldability. The research carried out by the authors of [14,15,16,17,18,19,20,21] confirms the above statement. In general, the results of these tests indicate that, among the considered materials, relatively satisfactory weldability is characterized by steels which CEV carbon equivalent does not exceed the value of 0.55. In practical terms, this condition significantly limits the possible range of sheets made of low-alloy wear-resistant steels offered by various manufacturers. In the case of the SSAB steelworks, these steels are classified according to the average Brinell hardness, i.e., Hardox 400, Hardox 450 and, to a limited extent, Hardox 500. For steels with a higher hardness index, e.g., Hardox 550, Hardox 600, Hardox Extreme, but also steels from other producers (e.g., XAR 600 [22]), the condition of metallurgical weldability is not met. The confirmation of the position quoted above is the location on the C-CEV diagram (Table 2 and Figure 1) of the actual chemical compositions of these steels in zone (III), i.e., high tendency for crack formation, regardless of welding conditions. Additionally, with regard to most of the materials quoted, there are large discrepancies between the producer’s data (PD in Figure 1) and the results of the author’s own research (OR in Figure 1). The shift of the CEV carbon equivalents of these steels into zone (III) proves to be a disadvantage of metallurgical weldability, which is also signaled by the operators of these materials. The most frequently mentioned problems in the context of weldability of Hardox 600 steel are the susceptibility to cold fracture after welding and quite frequent occurrence of the so-called delayed cracks (24–48 h after welding). The latter are observed both in the area of a welded joint and in the area of a base material that has been thermally treated (e.g., in a plasma cutting operation). It is worth mentioning that the occurrence of delayed fracture of Hardox 600 steel has been repeatedly noted by the author of this work—which is very important—not only in the material subjected to additional thermal treatments but also in thick sheets delivered directly from the steelworks.

With reference to the above statements, the aim of the research was to identify macro- and microscopic structure of welded joints of Hardox 600 steel, including delivery condition (immediately after welding) and a condition after thermal treatments. A very extensive set of thermal treatments, in terms of the conditions and parameters used, was aimed at causing structural changes, minimizing or completely eliminating the above-described unfavorable phenomena occurring during welding and post-welding heat treatment of this steel. It is also worth mentioning that similar tests were carried out on Hardox Extreme steel, which were presented in a separate study [25]. The selection of variants and parameters of individual technological operations was also carried out in terms of both obtaining high strength indices of a welded joint, as well as the highest possible resistance to abrasive wear. Therefore, in relation to Hardox 600 steel and its welded joints, additional tests of abrasive wear resistance under actual conditions of abrasive soil were carried out. However, due to the specificity of these studies and the amount of research material, their results require a separate study.

## 2. Research Material and Methodology

Steel sheets with approximate dimensions 1600 × 120 mm and a thickness of 10.0 mm were used for the tests, taken from a sheet of Hardox 600 steel (SSAB, Oxelösund, Sweden) characterized by the chemical composition listed in Table 2 and Table 3. Cutting the sheets to the required dimensions (Figure 2), taking into account the ridges intended for welding, was realized by means of a numerically controlled plasma system HyPerformance.

Welded joints were made using the SAW method (121), taking into account welding materials dedicated to high-strength low-alloy steels. Selected properties of the welding materials used are summarized in Table 4. The welding operation was performed using an ESAB A2 Mini Trac welding device with an ESAB LAE 800 power source (ESAB AB, Göteborg, Sweden). Hardox 600 steel sheets were joined with a double-sided weld (Figure 3), according to the following parameters that guarantee the correct penetration of the sheets:Weld type: BW (butt weld);Welding position: PA (flat);Electrode diameter: 3.0 mm;Arc voltage (weld layer: 1, 2): 35/35 V;Amperage (weld layer: 1, 2): 520/640 A;Polarity: DC (+);Welding rate (weld layer: 1, 2): 0.62/0.62 m/min.;Electrode wire: OK Autrod 13.43 (S3Ni2.5CrMo acc. to EN ISO 26304);Flux: OK Flux 10.62 (MgO, CaF_2_, Al_2_O_3_, SiO_2_);Pre-heating: no;Inter-pass temperature: ≤80 °C;Preparation of sheet edges (chamfering): no.

Before welding, the positions of the sheets were established by fixing them permanently to a backing material with tack welds. During welding, run-out sheets made of Hardox 600 steel were used. Figure 4, Figure 5 and Figure 6 show macroscopic images of the welded joint.

For the purpose of the tests, after welding, cuboid-shaped samples were cut out of the sheets (Figure 6). The machining operations were carried out with the use of a high-energy abrasive water stream and electro-erosion. In subsequent technological operations, some samples of welded joints were subjected to heat treatment, consisting of volumetric quenching in oil or water and low tempering. Before quenching, the samples were additionally normalizingly annealed. The thermal operations were performed in the FCF 12SHM/R gas-tight chamber furnaces by Czylok (CZYLOK Company, Jastrzębie-Zdrój, Poland) using a protective atmosphere of 99.95% argon inert gas. For quenching, Durixol W72 quenching oil with a kinematic viscosity of 21 mm^2^/s, heated to a temperature of 50 ± 5 °C, was used. A water bath consisted of deoxygenated water with a temperature not exceeding 30 °C. The detailed characteristics of the samples used for the tests and the parameters of the thermal treatments performed are presented in Table 5. For the required dimensional tolerances and surface roughness, at the final stage, all specimens were grinded (Figure 7).

Chemical composition analyses were performed with a spectral method using a Leco GDS500A (LECO Corporation, St. Joseph, MI, USA) glow discharge analyzer. During the analyzes, the following parameters were used to enable ionization of the inert gas: U = 1250 V, I = 40 mA, 99.999% argon. The obtained results were an arithmetic mean of at least five measurements.

Hardness measurements of the base material were made with the use of a universal hardness tester Zwick/Roel ZHU 187.5 (Zwick Roell Gruppe, Ulm, Germany) by the Brinell method, in accordance with the PN-EN ISO 6506-1:2014-12 standard. A cemented carbide ball with a diameter of 2.5 mm was used, with a load of 187.5 kgf (1838.7469 N) operating for 15 s. All measurements were made on samples subjected to prior micro-structure evaluation. Hardness measurements on selected cross-sections of welded joints were made in accordance with PN-EN ISO 6508-1:2016-10 by the Rockwell method (HRA), using the above-mentioned universal hardness tester, with a load of 60 kgf (588.399 N). The obtained hardness indices were converted to the Vickers scale in accordance with the PN-EN ISO 18265:2014-02 standard. Hardness measurements were carried out on the cross-sections of the samples in the state after welding and after the performed heat treatment. The places of the performed hardness distributions are schematically marked with lines A and B in Figure 3.

Macroscopic observations were made using a Nikon AZ100 (Nikon Corporation, Tokyo, Japan) multifunctional stereoscopic microscope. Observations of the micro-structure, on the other hand, were made with a Nikon Eclipse MA200 (Nikon Corporation, Tokyo, Japan) light microscope using magnifications in the range of 100–1000×. The tests were carried out on samples in a non-etching state and after etching with a 3% HNO_3_ in ethyl alcohol (Nital) and Adler’s reagent, in accordance with PN-H-04503:1961. Nikon DS-Fi2 (Nikon Corporation, Tokyo, Japan) digital cameras coupled with microscopes and Nikon’s NIS Elements (Nikon Corporation, Tokyo, Japan) software were used to record and analyze the recorded images.

Strength tests were conducted in ambient temperature on rectangular, proportional fivefold samples, as per PN-EN ISO 6892-1:2020-05. A material testing system MTS 810 (MTS Headquarters, 14000 Technology Drive, Eden Prairie, MN, USA) was used with an extensometer, measurement base length L_0_ = 25 mm. During the tests, the stretching rate was controlled by the stress increase rate. Next, the following were determined for each sample: Young’s modulus (E), elastic limit (R_p0.05_), yield strength (_Rp0.2_), tensile strength (R_m_), the percentage reduction of area (Z) and percentage elongation (A_5_) after tear. The final strength indexes were an arithmetic mean of the results obtained from at least five samples for each measuring point. Additionally, based on the test results obtained for individual samples, measurement errors in the form of standard deviation were calculated. Examples of samples subjected to strength tests are shown in Figure 8 and Figure 9.

Impact tests were performed for selected sets of samples of welded joints using the Charpy Zwick/Roell RPK300 hammer (Zwick Roell Gruppe, Ulm, Germany), using the initial energy value of 300 J, in accordance with the PN-EN ISO 148-1:2017-02 standard. The samples used in the tests were square with V-shaped notches. The method of sampling and the location of the notches in relation to the weld are schematically shown in Figure 3. Based on the results of tests carried out at temperatures of +20 °C and −40 °C, arithmetic means were determined—from at least 5 samples for each measuring point—and standard deviations of the obtained impact indexes. Then, on selected fracture surfaces, fractographic analyzes were performed using the JSM-6610A scanning microscope (SEM) by Jeol (JEOL Ltd., Tokyo, Japan). An accelerating voltage of 20 kV was used during the tests. Observations were made in the material contrast with the use of SE detectors. Based on the recorded images of the fracture surfaces, the percentage shares of plastic zones were calculated with the use of ImageJ free software ver. 1.52, developed for the needs of the National Institutes of Health.

## 3. Research Results

The main goal of all technological operations carried out in relation to Hardox 600 steel was to obtain a correct (free from incompatibility) welded joint, characterized at the same time by the entire heat-affected zone, with structural and strength properties as close as possible to the properties of the base material. The implementation of the set goal was based on the use of post-welding heat treatment, consisting of volumetric quenching in a water or oil bath, and then tempering, using various temperature–time variants. In selected cases, the quenching process was preceded by application of normalizing annealing. The selection of the optimal austenitizing temperature was made taking into account an actual chemical composition of the base material (Table 2 and Table 3) and based on the weld metal composition obtained as a result of welding within the entire cross-section of the weld (Figure 3 and Table 5).

When selecting appropriate parameters of thermal treatments, theoretical CCT and TTT charts were also used, obtained by computer simulation with the use of JMatPro software, included in [4]. All the diagrams used and the detailed parameters of the performed thermal treatments are presented in Table 6. This table also summarizes the results of strength and impact tests. In the case when the results of selected thermal treatments raised serious doubts, no impact tests were undertaken. A detailed explanation of this issue is presented in the further part of the study. Table 6 also presents the results of mechanical properties tests of the base material subjected to the selected (optimal) variant of heat treatment. Due to the fact that, along with the entire heat-affected zone of the welded joint Hardox 600, the zone of the base material is also subject to volumetric heat treatment, it seems reasonable to determine the impact of these treatments on the selected mechanical indicators of the welded sheet. Figure 10 and Figure 11 show hardness changes along the entire cross-section of the welded joint, corresponding to the individual thermal treatments listed in Table 6.

### 3.1. Static Tensile Test and Hardness Measurements

Strength tests of the welded joints of Hardox 600 steels have shown that, in the state immediately after welding (process No. 1 in Table 6), they have an average tensile strength of R_m_ = 879 MPa. In the context of data provided by the producer (R_m_ ≥ 2000 MPa, Table 1), as well as those determined and presented in [4] and Table 1 of the actual mechanical parameters of the tested steel, it can be indicated that the obtained index of the static strength of the entire welded joint is only 41–44% of the strength of the base material. It is also worth emphasizing that the discussed strength parameter is obtained at a relatively low level of relative elongation (A_5_) and percentage reduction of area (Z), which, in this case, is also associated with a low level of impact toughness, reaching already at ambient temperature a value below a brittleness threshold, often defined by the value 35 J/cm^2^ [27]. The above-mentioned observations lead to the conclusion that properties of the welded joint obtained in the SAW technology in the delivered condition do not meet the basic strength requirements, which, in each case, should be related to the properties of the base material. Additionally, the obtained low impact toughness indices clearly indicate the possibility of fragility fracture, both during operation and at the stage of preparation and implementation of the welded structure (structural weldability). Therefore, according to the author of this study, what is emphasized by the producer of Hardox steel frequently, as well as by many other producers of high-strength steels resistant to abrasion, the classification of this material group in terms of weldability, should be changed (extended). Otherwise, the use of welding processes for them without additional post-operative procedures within the entire heat-affected zone—in relation to the base material—leads to a reduction in mechanical properties, deterioration of resistance to abrasive wear, and increased potential of brittle fracture. The use of quenching and tempering operations after welding (process No. 2 in Table 6) against Hardox 600 also did not produce satisfactory results. Subjecting the entire heat-affected zone to the above-mentioned treatments leads to the reconstruction of hardness levels within the welded joint (Figure 10 and Figure 11); nevertheless, despite a significant increase in the elastic limit and plastic extension, in virtually every test of the test procedure, it was subjected to brittle fracture before reaching the theoretical tensile strength. The immediate reasons for this were numerous discontinuities in the micro-structure observed throughout a fusion line, impossible to be leveled without additional thermal treatments prior to quenching.

Taking into account the previously described problems of reduced mechanical and functional properties of welded joints of the analyzed steel, it seems reasonable to use before-quenching, normalizing annealing operations or technological operations to enable regulated cooling after austenitizing. From a scientific point of view, the use of uniform annealing treatments should also be considered. Such treatment, however, most often requires long-term isothermal annealing in conditions of a strongly overheated austenitic phase. In addition, it largely eliminates the influence of multiple re-crystallization and phase crushing of steel sheets obtained in the course of thermomechanical rolling, and therefore makes it impossible—from a practical approach to the problem—to restore structural and strength properties of the base material. Table 6 presents the results of strength tests of welded joints of Hardox 600 steel, subjected to normalizing annealing, carried out at the temperature of 850 °C (process No. 3) and 900 °C (process No. 4). In both cases, obtained mechanical properties and hardness distributions (Figure 10 and Figure 11) were almost identical and corresponded very well with the parameters of the welded joint as delivered. In relation to the latter, the normalization operations clearly led to homogenization of the micro-structure of entire heat-affected zone and the base material, at the same time translating into an increase in plastic properties by about 19–40% in the case of elongation (A_5_) and 80–83% in the case of a percentage reduction of the area (Z). Due to the obtained favorable mechanical and structural properties (considered in a separate part of the manuscript), it seems most justified to analyze the subsequent variants of thermal treatments performed, including the normalizing annealing operation. It is also worth mentioning that, in the discussed case, no impact tests were carried out. This is due to the fact that, in the case of Hardox 600 steel, the use of normalization as the final heat treatment leads to a reduction in fracture toughness, resulting from high activity of precipitation processes during relatively slow cooling. Therefore, according to the author, the use of normalizing annealing treatments as a separate process should not be considered at all. On the other hand, very favorable phenomena in the case of the application of technological operation in question are observed in complex thermal treatments of welded joints, including final quenching and possible tempering.

On the basis of the above considerations, Table 6 summarizes the results of research on the mechanical properties of several temperature–time variants of complex heat treatments of a welded joint of Hardox 600 steel, taking into account the normalizing annealing operation (processes 4–11). Figure 10 and Figure 11 also present hardness changes for all heat treatment schemes listed in Table 6. On the basis of the obtained test results, it can be concluded that the highest level of strength was obtained with the use of volumetric quenching in water or oil and then tempering at the temperature of 100 °C (processes 5, 9, 11). In all three cases, the obtained value of static strength exceeded 1800 MPa while maintaining satisfactory plastic properties. Regardless of this, it is worth pointing out that the application of a higher austenitizing temperature before quenching in a water bath (process 11) increases the plastic extension of the welded joint. However, it should be noted that obtaining high strength indices of the samples, heat-treated in accordance with the presented diagrams, depends on carrying out the normalizing annealing process in a slightly lower—compared to quenching—temperature range. Otherwise, the results will not be satisfactory (process 10). This is due to the fact that, during the normalization, as well as in an initial cooling stage after this process, complex precipitation processes occur in steel, resulting in the depletion of austenite in carbon and, thus, a lower degree of supersaturation of a solid solution with this element during the quenching operation. A detailed explanation of this phenomenon required numerous studies with the use of X-ray electron microscopy (TEM), which, due to the very extensive nature of the discussed issues, require a separate study. It is worth adding, however, that a similar effect is also observed in other micro-alloyed steels with chemical compositions and functional properties similar to Hardox 600 steel. On the other hand, referring to the value of the required quenching temperature, the author’s experience shows that, in the normalizing temperature range 850–900 °C, increasing the quenching temperature by 50 °C allows for obtaining of optimal heat treatment conditions for the welded joint of the tested steel, ensuring high mechanical properties.

The above considerations regarding the possible strength properties of a welded joint of Hardox 600 steel, subjected to comprehensive heat treatments, should be additionally supplemented with the aspect of increasing the susceptibility to fracture within the base material, observed during numerous research tests. Due to the metallurgical properties of Hardox 600 steel (very high hardenability) and the technology of producing sheets from this steel (thermomechanical rolling), in most cases, the use of a water bath during quenching led to cracks in the axis of the sheet running in the direction of its plastic processing. The appearance of cracks was most often observed after 24–48 h from the quenching process completion. It should also be added that this process was not dependent on the austenitizing temperature before quenching, the tempering temperature or the time interval between quenching and tempering. The described phenomenon was dictated only by a cooling intensity factor during quenching treatments. Therefore, the optimal—from the technological point of view—scheme of thermal treatments in relation to the analyzed welded joint of Hardox 600 steel should be considered: normalization at 850 °C, quenching in oil (heated to a temperature of ~50 °C) and tempering (stress relieving) at 100 °C (150 °C) (process No. 9 in Table 6). Therefore, further considerations on the structural properties of the welded joint of the tested steel were limited to a delivery condition and to a condition after quenching in oil. It is also worth adding that the application of the above heat treatment scheme also allows—apart from the optimal parameters of the heat-affected zone—restoration of the structural and strength properties of the base material. Table 6 (the process marked as BM) presents the results of testing the strength and impact properties of the base material samples made of Hardox 600 steel, subjected to heat treatment analogous to those described above. In terms of mechanical properties, heat-treated samples of the base material, apart from a slightly lowered yield strength (R_p0.2_), showed almost identical parameters as samples taken from Hardox 600 steel in the smelting condition (Table 1). Accordingly, the heat treatment scheme No. 9 can be considered optimal, both in relation to the entire heat-affected zone of the welded joint and in relation to the base material.

### 3.2. The Results of Microscopic Examinations

Figure 12 shows an image of the micro-structure of Hardox 600 steel before the welding processes—as delivered. In such a state, the tested steel is characterized by a fine-lath tempering martensite structure with numerous features of a banding structure. These bands are created by the quenching martensite and are the result of thermo–mechanical rolling at the stage of producing Hardox steel sheets. The martensitic structure obtained in the course of metallurgical procedures is characterized by fine needle-like and fine-lath morphology, locally—within the boundaries of the former austenite grains—arranged in the form of characteristic blocks and packages.

Figure 13, Figure 14, Figure 15, Figure 16, Figure 17, Figure 18, Figure 19, Figure 20, Figure 21, Figure 22, Figure 23, Figure 24, Figure 25, Figure 26, Figure 27, Figure 28, Figure 29, Figure 30, Figure 31, Figure 32, Figure 33, Figure 34, Figure 35 and Figure 36 show macroscopic images and an overview of characteristic micro-structures of the welded joint of Hardox 600 steel, both as delivered and after selected heat treatment. Immediately after welding (Figure 13, Figure 14, Figure 15, Figure 16, Figure 17, Figure 18, Figure 19 and Figure 20), the entire area of the wide heat-affected zone is characterized by very diverse structural changes. The macroscopic image (Figure 13) shows selected zones of the tested steel welded joint, which were subjected to a broader micro-structural analysis in the further course of this study. In the zone of the weld metal, which was made in the first place (area WM1 in Figure 13), the structure of non-equilibrium ferrite is observed in places with precipitations of the quenching troostite (Figure 14). On the other hand, no clearly visible pearlite colonies are observed, the presence of which may result from the carbon content in this zone, i.e., 0.30% (point X in Figure 3 and Table 5). In the zones with increased carbon content, band-like precipitation of quenching martensite is observed, proceeding in the direction of crystallization during solidification of the weld metal. Due to the lack of clear features of the dendritic structure, it can be indicated that the considered zone of the welded joint underwent re-crystallization as a result of thermal processes occurring during welding the second layer of the weld. As a result of this treatment, in the zone of the weld metal marked as WM2 in Figure 13, features of the dendritic structure with numerous precipitates of the quenching structures can be clearly observed (Figure 15). In general, it consists of quench sorbite—locally, with the characteristics of upper bainite with block ferrite precipitates—as well as numerous regions of martensite or lower bainite, crystallized due to supercritical cooling conditions. In this zone, it is also possible to indicate the presence of wide-angle migration boundaries (Figure 15b), the presence of which indicates a zonal excess of heat during crystallization from the liquid phase or a reheating cycle, i.e., during the execution of successive layers or weld beads [28]. It can be concluded that the structural changes observed in the WM2 zone significantly differ from those observed in the WM1 zone, despite the presence of very similar chemical compositions of the weld metal in both zones (point Y in Figure 3 and Table 5). The above considerations are confirmed by a construction of a transition zone between WM1 and WM2 (point Z in Figure 3 and Table 5). The structural structure of the very clearly delineated zone marked as WM3 in Figure 13 consists mainly of quenching sorbite—formed along the fusion line of both layers of the weld (Figure 16). Moreover, in the discussed zone, one can also observe the presence of a martensitic or martensitic–bainitic structure, as well as few colonies of the quenching troostite, most often coexisting with the martensitic areas.

Figure 17 shows structural changes within the fusion line occurring during the execution of the first layer of the weld (FL1 area in Figure 13). In the case of the above zone, it can be stated that it is dominated by a sorbitic structure with distinct features of post-martensitic orientation. This state of affairs indicates thermal processes that are characteristic of tempering within a fusion line. The regions of martensite or lower bainite in the discussed zone are observed relatively sparse, which seems to confirm the earlier statement. On the other hand, the presence of bainitic ferrite plates (designation B in Figure 17c,d), thickness of which is 3–5 µm (length ~60 µm), is slightly more numerous. The presence and large size of this phase indicates local (heterogeneous) growth of austenite during welding and also indicates a possibility that, during crystallization of this zone, phase transformations could take place in isothermal conditions. The type of structure described above, in relation to the heat-affected zone of welded joints, most often manifests itself in a significantly reduced brittleness threshold, observed even at ambient temperature.

In the case of the zone of the tested welded joint marked as FL2 in Figure 13, it can be stated that it consists predominantly of a ferritic–sorbitic structure and a strip-like arrangement—in the direction of crystallization—of quenching martensite (Figure 18). Secretions of lower bainite and troostite are also locally observed. The lack of clear features of the post-martensitic orientation of the sorbite formed and a fairly significant amount of block ferrite allow us to conclude that the pattern of changes taking place on the fusion line during the execution of the second layer of the weld was characteristic of those observed during partial or isothermal quenching. The described type of construction on the fusion line may also indirectly result from the dendritic structure remaining after directional crystallization, which, in the FCC or BCC crystal lattices, most often runs along the <100> direction [28]. High rate of crystal growth is also associated with a high dendritic segregation index of the chemical composition. Hence, very often on the fusion line of welded joints and surfaced layers there are observed structures indicating lower carbon content than it would result from the chemical composition of the base material and an additional material used. Regardless of the above considerations, due to the presence of migrated grain boundaries (Figure 18d), it can be concluded that there is a significant excess of heat resulting from the supplied linear energy during welding. This state of affairs was dictated by the welding technology used and the thickness of the Hardox 600 sheet used. It is also worth mentioning that, due to very high martensitic hardenability of the welded steel and the presence of alloy micro-additives preventing the growth of austenite grain, no clear features of Widmanstätten structure are observed in the fusion zone. Figure 19 presents a summary of structural changes occurring in selected areas of the heat-affected zone, located outside the weld metal and the fusion lines of individual layers of the weld. In the area marked as FGH in Figure 13, the presence of different structures, characteristic of the quenching process or—directly in relation to Hardox 600—normalization, was demonstrated (Figure 19b–d). They include significant amounts of martensite or lower bainite, distributed in strips in the direction of thermoplastic processing of the sheet, as well as areas of quenching sorbite. The presence of a few troostite colonies, areas of upper bainite and structurally free ferrite can also be distinguished. Figure 19e,f presents the inter-critical heat-affected zone of the welded joint, marked as ICH in Figure 13. Unlike the FGH zone, the ICH zone is characterized by the presence of structural changes resulting from a tempering process. In general, it can be concluded that it includes a structure characteristic of tempering sorbite with sparse troostite colonies. On the other hand, the structure of the base material (area BM in Figure 13) can be characterized as tempering martensite with areas of quenching martensite (Figure 20). Moreover, in the structure of the zone in question, slight banding features, resulting from steel production process, can be indicated.

The above-described structural changes within the entire welded joint of Hardox 600 steel in the delivered condition lead to the conclusion that shaping mechanical and functional properties of this material, resulting only from the technology, welding parameters and conditions used, can be implemented to a very limited extent. The technological trials undertaken, consisting in the re-quenching of selected fragments or the entire heat-affected zone, without the use of other thermal treatments, did not produce satisfactory results. Despite the reproduction of high hardness levels and selected strength parameters, it was not possible to obtain optimal plastic properties and, above all, impact toughness. Therefore, according to the author, implementation of the above objectives is not possible without standardizing structural changes within the selected zones of the welded joint prior to subsequent thermal treatments, e.g., volumetric or surface quenching.

Figure 21 shows a macroscopic image of the considered welded joint after performed normalizing annealing. In such a state of heat treatment (immediately after welding), the same zones of the welded joint were subjected to structural analysis as in the case of the delivered condition. Nevertheless, due to high degree of uniformity of the structure in the area between the FL1/FL2 fusion lines and the base material BM (Figure 21), in relation to this zone, the tests were generally carried out for the entire heat-affected zone (HAZ designation in Figure 21).

In the zones of the weld metal, marked as WM1 and WM2 in Figure 21 (in terms of chemical composition, points X and Y in Figure 3 and Table 5), the normalization procedure caused very similar structural changes (Figure 22 and Figure 23). In both cases, the obtained structure was characteristic of incomplete quenching treatments. It consisted mainly of non-equilibrium ferrite grains, locally with the features of upper bainite, with a significant volume of a quenching sorbite structure. The presence of a martensitic structure is observed in the carbon-enriched zones, in the form of micro-structural bands running in the direction of weld crystallization during welding. In a few zones, the presence of the quenching troostite structure can also be indicated.

The transition zone between the materials of the weld metal WM1 and WM2 (in terms of chemical composition, point Z in Figure 3 and Table 5), marked as WM3 in Figure 24, in terms of the phase structure, shows great similarities to the zones discussed above. On a very poorly outlined fusion line (Figure 24a), the presence of a non-equilibrium ferrite structure as well as sorbite and quenching troostite (Figure 24b) can be indicated. In contrast to WM1/WM2 weld deposit zones, in the WM3 zone, band precipitation of martensitic areas occurs rather sporadically. On the fusion line of this zone, the presence of quenching martensite shows an island character, which proves the re-crystallization of this area as a result of welding and the performed normalizing annealing. It is also worth pointing out that the use of normalization after the welding process does not remove the banding features of the material, formed in metallurgical conditions in the course of thermomechanical rolling.

Figure 25 and Figure 26 show images of the micro-structure of a welded joint in the normalized state, i.e., on the fusion lines of the weld No. 1 and 2 with the base material, marked as FL1 and FL2, respectively, in Figure 21. In terms of structural, both zones show quite significant differences from each other. The micro-structure of the rather poorly indicated FL1 fusion line (Figure 25) can be described as characteristic of isothermal quenching. It mainly consists of sorbite and quenching troostite with relatively few martensitic regions. In the discussed area, it is also possible to indicate the presence of large-size bainitic ferrite plates (not re-crystallized after welding), the length of which exceed 50 μm (Figure 25b). The mechanism of the formation of this type of structure was outlined in the discussion of structures immediately after welding. It is worth adding, however, that structures of this type are most often much more resistant to tempering processes than typical quenching structures. Therefore, their presence is very often observed in the micro-structure of welded joints, even after normalization procedures have been performed. Moreover, the presence of very few areas with Widmanstätten structure features (designation W in Figure 25b) may also indicate high overheating during welding of the considered zone.

In the case of the zone marked as FL2 (Figure 26), it can be stated that in the normalized state it is characteristic of the state after incomplete quenching, and, in terms of structure, it is very similar to the WM3 zone. The zone of weakly outlined fusion line (symbolically marked in Figure 26a) mainly consists of non-equilibrium ferrite grains, as well as quenching sorbite and quenching troostite. Directly on the fusion line, practically no precipitation of the martensitic structure, which has a much larger share in the zones deeper towards the weld metal, is observed. The discussed type of structure primarily results from the significant dendritic segregation of the chemical composition during welding.

The heat-affected zone subjected to normalization and the zone of the base material, marked in Figure 21 as HAZ and BM, respectively, are very similar to each other. They are characterized by a fairly diversified structure, characteristic of a hardened state isothermally (Figure 27 and Figure 28). In both cases, their structure consisted mainly of quenching sorbite and quenching troostite, as well as band precipitation of the martensitic structure, in many cases transforming into a bainitic structure (Figure 27b and Figure 28b). It is worth noting that, both in the HAZ and BM zones, precipitations of the structurally free ferritic phase practically do not occur. This indicates optimal cooling conditions for both zones, resulting from similar chemical composition and the applied heat treatment parameters.

Figure 29 shows a macroscopic image of a welded joint of Hardox 600 steel after comprehensive heat treatment, according to the diagram No. 9 shown in Table 6. As in the previous cases (as delivered and normalized), the welded joint was subjected to structural tests in selected zones, symbolically marked in the macro-photograph, below. Figure 30, Figure 31, Figure 32, Figure 33, Figure 34, Figure 35 and Figure 36 present an overview of the structural structure in the characteristic zones of the tested welded joint. Regardless of the zone under consideration, it can be stated that the implemented thermal treatments caused very similar structural changes in the entire area of the butt weld. In practically every case under consideration, the welded joint was characterized by a fine-lathed tempering martensite structure with areas of quenching martensite. The above statement seems to be correct both for the zone of the base material and the weld metal and for the structural properties observed on the fusion lines.

The weld metal zones marked as WM1 and WM2 presented in Figure 30 and Figure 31 show almost identical structural features, which is largely due to the same chemical composition in these areas (points X and Y in Figure 3 and Table 5). In both cases, it consists of fine-lath tempering martensite with banded quenching martensite—untempered martensite. These bands run in the direction of the primary crystallization of the weld, caused by the welding process. Locally, apart from these bands—in the zones depleted in carbon—martensite with a packet and block structure is observed, characterized by a low variability of the crystallographic orientation of individual packets within the boundaries of the former austenite grains (arrows A in Figure 31b). This condition is typical of hardened low-carbon steels.

A slightly larger volume of the block martensite structure can be indicated on the fusion line between the weld metal materials (Figure 32). Almost along the entire length of the very poorly defined fusion line, marked as WM3 in Figure 29 (in terms of chemical composition, point Z in Figure 3 and Table 5), the presence of low-carbon, fine-lath block martensite is observed (MB designation in Figure 32b). This state of affairs is dictated by the depletion of this area in carbon, resulting in reduced hardenability and resistance to tempering processes. Hence, in sporadic cases, the presence of highly tempered structures with post-martensitic orientation is noted (marking MT in Figure 32b). It is worth mentioning that removal of the described changes would require additional homogenizing annealing, which, due to the properties of the base material, is inadvisable, as well as technologically difficult and economically unjustified.

Referring to the structural properties recorded on individual FL1 and FL2 fusion lines of the welded joint under consideration (Figure 33 and Figure 34), it should be noted that the comprehensive heat treatment procedures carried out led to an almost complete homogenization of the structure in these zones. As a result, the structure of fine-lathed, medium-carbon tempering martensite with few zones of quenching martensite was obtained (marking MQ in Figure 33). It is also worth adding that there are relatively few areas of the martensite structure with block morphology in this case, which indicates a similar percentage of carbon in these areas.

Micro-structure of the heat-affected zone and the base material, marked respectively as HAZ and BM in Figure 29, showed no significant differences as a result of the thermal treatments performed. In both cases, a similar structure is observed, consisting of tempering martensite with numerous bands of unremoved martensite structure (MQ designation in Figure 35 and Figure 36). The presence of these bands is a direct result of producing Hardox 600 sheets technology—thermomechanical rolling—which were not “removed” during the applied post-welding thermal treatments. It is also worth mentioning here that the structural properties of the zones in question, especially the BM zone shown in Figure 36, are very similar to the properties of the base material supplied directly from the steelworks, shown in Figure 12.

### 3.3. Results of the Fractographic Analysis

Figure 37 presents macroscopic images of the fracture surfaces of welded joints representative samples, subjected to impact toughness tests at +20 °C and −40 °C. As a result of the fractographic analysis, very significant differences in the overall structure of fractures between the analyzed series of samples can be identified. Regardless of the test temperature, the fractures of welded joints samples as delivered (immediately after welding) do not show a significant share of plastic side zones. At the temperature of +20 °C, the share of the plastic zone occurring only locally does not exceed 2% of the total fracture area (frame D1 in Figure 37). In the case of samples subjected to the impact test at −40 °C, these shares do not exceed 1% (frame D3 in Figure 37); therefore, their presence can be described as marginal. The above-described state of affairs proves the expenditure of very little energy during the formation of fractures, which also confirms the obtained very low impact indexes of the welded joint in the delivered condition. In the case of fractures of welded joint samples subjected to complex heat treatment, in the entire temperature range of impact tests, they show significant shares of plastic side zones. At a temperature of +20 °C, these shares can be estimated at 20% (frame D2 in Figure 37) and about 12% for the Charpy test carried out at 40 °C (frame D4 in Figure 37). The clear presence of relatively wide plastic side zones on the fracture surfaces of heat-treated samples indicates formation of fractures over a longer period of time, with the simultaneous expenditure of considerable energy. The above statement seems to be justified by the values of impact indices obtained (Table 6). In addition, the macroscopic analysis performed showed that the fractures of the samples in the delivered condition are characterized by a different topography, resulting from the presence of fractures almost on the entire surface (except for the fracture zones marked with frames C1 and C3 in Figure 37), a coarse-grained structure on the fusion line. On the other hand, of the welded joint surfaces of the fractures subjected to additional heat treatment can be considered fine-grained, homogeneous, characterized by roughness, running in the direction of crystallization during welding. In order to identify the structure of the fracture surfaces individual zones in question, all samples were subjected to further examination using SEM electron microscopy.

Figure 38 and Figure 39 present images of the fracture surfaces in the area under the mechanical notch—frame A in Figure 37. As a result of the fractographic studies, it can be concluded that, in terms of side fractures of the welded joint in the delivered condition (not heat-treated), there are brittle transcrystalline fractures. This statement applies to both samples after the Charpy test at −40 °C and at ambient temperature (Figure 38a,c and Figure 39a,c). Fractures of this type arise as a result of fracture mechanism occurring along specific crystallographic planes—cleavage planes—depending on the type of material (structure) and loading conditions. In metallic materials with a spatially centered regular lattice, these are usually the {100} [29] planes in which the plastic deformation is the smallest. It is worth noting that the discussed type of fracture is most often initiated by various types of blocking mechanisms (e.g., defects in the crystallographic structure), but it can also nucleate at the grain boundaries. Therefore, a cleavable fracture often exhibits a very varied surface topography with many distinctive features. Figure 39 shows selected fractures of the welded joint in the delivered condition. There are sets of numerous steps (S) located on large (in terms of size) facets (F). These steps create a characteristic “river pattern” (RP), often cutting across whole grains. The presence of steps proves that the fracture course did not occur only in one crystallographic plane but repeatedly passed to the adjacent planes by cutting or secondary fracture of the dividing walls. The steps are seen where the front of the fracture hits the helical dislocation, while their height is related to the amount of energy absorbed during fracture—reduction of the brittleness threshold [30]. Steps with a fairly flat course, characterized by a very extensive system of “river pattern”, dominate on the surfaces of the studied fractures. These features indicate a small amount of energy absorbed during the crack formation, which is also confirmed by the impact toughness indexes obtained (Table 6, No. 1). In the context of the results of structural studies presented in this manuscript, it is also worth referring to the observed correlation between the fracture morphology of unheated samples and their micro-structural features. In iron alloys with a structure different from that of martensite or lower bainite, the course of splitting fracture is determined only by a grain size of the former austenite [29,30]. Thus, it does not depend on the type and number of secondary phases and mainly depends on the crystallographic orientation of individual grains or micro-structure blocks. This is due to the fact that the development of cleavage fracture is inhibited at the grain boundaries, which is reflected in the formation of the set of the steps described above. The precipitation of foreign phases inside the grains cause a temporary slowdown (stop) of the fracture process, initiating new systems of “river pattern”. On the other hand, the precipitation of foreign phases at the grain boundaries is attributed to the occurrence of inter-crystalline cracks (IC designation in Figure 39a,c). Segregation of the chemical composition may have a similar effect on the nature of fracture. In addition to the features discussed above, the so-called “tongues” (T) are also noted in the fracture in question. According to [29], their presence is related to a local plastic deformation during fracture at the boundaries of micro-structure blocks as a result of encountering a twin boundary formed in front of the fracture front. Therefore, as a result of this process, the fracture front “flows around” (bypasses) these zones without going beyond the original plane of its course—in steels with A3 lattice—the {112} plane. Therefore, it can be concluded that the observed morphological features of the welded joint fractures in the delivered condition confirm the identification of phases and structure components (ferrite, sorbite, troostite, upper bainite and martensite regions).

In the case of samples of a welded joint subjected to heat treatment, in the area under the mechanical notch—frames A2 and A4 in Figure 37—at both impact test temperatures, the nature of the fractures can be described as mixed (plastic–brittle or quasi-cleavable)—Figure 38b,d and Figure 39b,d. On the surfaces of the fractures, a dimple rapture of various sizes of voids dominates. Fractures of this type are formed by coalescence of micro-voids —already existing in the material or formed and grown during loading—up to the point of plastic decohesion. It is observed that micro-voids may form at the boundaries of non-metallic inclusions or secondary phase precipitation with the material matrix at the boundaries of these phases or as a result of mechanical destruction of inclusions, which locally lead to stress accumulation [29,31]. In most cases, at the bottom of the cavities (voids), both complete and damaged foreign phases are observed, as well as cavities after their removal. According to the discussed mechanism of dimple rapture formation, it seems logical to say that the size of the voids depends on the type, shape, number and distribution of inclusions facilitating the nucleation of micro-voids, as well as on the grain size, structure and plastic properties of the material. It can also be concluded from [31] that the inhomogeneous distribution of inclusions as well as nucleation and growth of isolated micro-areas (micro-voids) during the initial stage of loading the sample lead to the formation of a fracture with a different size and arrangement of voids. In the case where the nucleation of the micro-voids takes place at the grain boundaries of the material’s micro-structure, the resulting fracture will be inter-crystalline.

It is also worth noting that, in metals of high chemical purity or metal alloys subjected only to supersaturation treatments (not aged, not tempered), characterized by the absence or a small number of secondary phases undissolved in solid solution, the presence of deep voids can only occasionally be observed. In such a case, the fracture structure is usually dominated by shallow recesses growing in lateral directions. From the point of view of a loading method, it is also possible to distinguish between even voids—uniaxial tension—and elongated voids with a parabolic scale-shaped contour, which are caused by shear or unequal tension. In such a case, formation of the characteristic “fish scale” structure may take place without the participation of foreign phase inclusions [29]. Additionally, depending on shapes and types of the discussed void structure, the following types of fractures can be distinguished: normal plastic fracture (uniform voids), when the direction of stress is approximately normal to the scrap surface and the course of crack development is relatively slow, sheer plastic fracture (elongated voids with a parabolic contour) and a plastic fracture by tear, characterized by elongated voids with a parabolic end, facing the direction of the fracture development [29,31]. Therefore, the presence of the cited “fish scale” structure may result from different nucleation rates and crack development in individual fracture zones.

Figure 39b,d show the characteristic fracture zones of the analyzed heat-treated samples. The fractures obtained at both ambient and low temperature are characterized by the presence of voids of very different sizes. The discussed feature is particularly visible in the fracture zones with high plastic deformation (DF), where the presence of extremely small voids (V) is observed. The remaining areas are dominated by larger, shallow voids in which no phase precipitation is observed. Such a situation results directly from the state of heat treatment, which, in the final stage, included tempering operation at a fairly low temperature (100 °C), i.e., much below the temperature favorable to the precipitation processes. In some areas of the revealed fracture surfaces—especially at −40 °C—single non-metallic inclusions (I) are visible, which are fine slags formed directly in the welding process, most likely resulting from the presence of corrosion products on the ridges of the welded sheet. On the fracture surfaces, one can also observe a characteristic “fish scale” structure (FS) approximately running in the direction of the crack development (especially at ambient temperature). This is confirmed by the fact that the fracture process is initiated by plastic deformation within the voids, while the fracture itself takes place similarly to brittle fracture by cutting the walls separating them. In addition, the presence of the so-called “tear ridges” (TR) that indicate the presence of a quasi-fracture. Such a fracture is created by joining many local fractures, usually formed on the same crystallographic plane [29], resulting in material’s decohesion, preceded by a strong plastic deformation. The presence of steps can also be indicated on the considered fracture surfaces—in brittle zones (BF). Nevertheless, their course is rather flat, without formation of the characteristic structure of “river pattern”, so it can be concluded that their contribution to energy absorption during destruction was marginal.

Figure 40 and Figure 41 show images of the fracture surfaces in the central zone of the impact specimens—frame B in Figure 37. The fractures of the samples as delivered, for both test temperatures, are cleavable fractures with small areas of plastic fractures (Figure 40a,c and Figure 41a,c). The presence of plastic zones is particularly observed on the sample tested at +20 °C. As a result of the fractographic analysis, it was shown that it is characterized by a very extensive structure, with various features of the surface topography—Figure 41a. There are numerous fractures in a form of deep craters, and the bottom of which is dominated by a transcrystalline fracture with rather deep steps (S), forming the characteristic of “river pattern” (RP). This proves the expense of considerable energy during fracture and the course of the fracture front along many crystallographic planes, orientation of which, in this case, depended on the grain size of the former austenite. On the other hand, formation of a fairly extensive system of “river pattern” most likely results from the inhibition of a fracture front due to the presence of foreign phases, separated inside grains or micro-structure blocks. The same can be said for the so-called “tongues” (T) and “tear ridges” (TR). It is worth noting that the simultaneous presence of steps, tongues and tear ridges at the fracture allows us to characterize the fracture as quasi-cleavable [29]. It is also worth noting clear presence of ductile fractures’ zones (DF) on the side walls of the formed craters. These zones are created by a characteristic dimple rapture, occurring in planes approximately consistent with the action of main stress. The presence of large voids (V), with a distinct elliptical contour, allows us to state that the decohesion of the sample material was preceded by a large plastic deformation and the coalescence of micro-voids occurring in the zones of clearly fragmented micro-structure. The fracture characteristics described above indicate the formation of a fracture under conditions of complex stress.

In the case of samples’ fractures subjected to heat treatment after welding, taken from the central zone—frames B2 and B4 in Figure 37, it can be concluded that they show similar features of a mixed fracture, i.e., quasi-cleavage with many plastic areas. This statement applies to both impact test temperatures (Figure 40b,d and Figure 41b,d). In both cases, the plastic zones form transcrystalline fractures with a dimple structure. In the case of the fracture obtained at ambient temperature, small-sized voids (V) with relatively regular shapes dominate. Additionally, clearly visible “fish scale” structure (FS) running approximately in the direction normal to the action of the main stress (in the direction of fracture) may indicate a failure mechanism due to unequal tension and shear. Only sporadic presence of non-metallic inclusions (I), mainly located inside larger voids and at the boundaries of the micro-structure blocks, indicates that the initiation of micropore formation took place without the participation of foreign phase inclusions. Lowering the impact test temperature to −40 °C resulted in a clear reduction in the fracture surface of the analyzed sample in the proportion of plastic zones (Figure 41d) dominated by the presence of voids (V) of various sizes, largely characterized by a parabolic outline.

In larger voids, as in the case of the sample tested at ambient temperature, the presence of non-metallic inclusions (I) is observed sporadically. On the fracture surface, the “fish scale” (FS) structure feature is also observed, occurring mainly in the zone of finer grain structure. Its volume share in relation to positive test temperature is much smaller, which indicates the course of fracture in a shorter time. With regard to both fractures of the heat-treated samples, it is also worth pointing out that the cleavage zones of their fractures do not show distinct features of the structure in the form of steps, forming the “river” pattern. The steps (S) present, rather, run on the boundaries of micro-structure blocks, and in special cases they can also be classified as tear ridges (TR). This situation indicates that the absorption of energy during fracture in the central part of the samples took place mainly through nucleation and coalescence of micro-voids.

The fracture zones of the tested samples—frames “C” in Figure 37—are shown in Figure 42 and Figure 43. The fractures of the samples in the delivered condition, tested at ambient temperature, can be defined as mixed fractures (plastic-brittle) with significant shares of plastic zones (Figure 42a and Figure 43a). Plastic zones are characterized by a void structure, with a fairly even arrangement and size of the voids, overlapping in the form of “scales” (Figure 42a). The features of the fracture structure, clearly visible in Figure 42a, indicate that the plastic zone of the fracture was created both by equiaxial tension and by the so-called tearing, resulting in the formation of voids with a parabolic contour facing the direction of the fracture development. It can also be indicated that the sources of micro-voids formation, which, as a result, of coalescence formed voids of larger sizes, were numerous inclusions of foreign phases (I). The presence of inclusions is recorded both at the bottom and on the side walls formed by the crater voids. The cleavage zones of the fracture in question show the presence of numerous steps (S), converging radially to the axis of the crater, creating at the same time a characteristic “river pattern” (RP). The height of the steps can be described as quite varied, in places flat and slightly outlined, and locally quite high—merging with tear ridges (TR). This state of affairs indicates that, in the fracture zone of the sample, considerable energy was expended during the fracture formation. It is worth emphasizing that the above statement can be applied both to plastic and quasi-cleavage zones of the fracture. On the other hand, the fracture of the same sample, tested at −40 °C, shows a quite similar structure to the fracture discussed earlier, but it shows much smaller shares of plastic zones (Figure 42c and Figure 43c).

Thus, it is dominated by a cleavage fracture with a relatively small number of rather high steps (S). The system of so-called “rivers” (RP) is observed rather sporadically. This proves the course of the fracture in a relatively short time, which significantly hindered the formation of twin boundaries at the front of the fracture. It is also worth pointing to the presence of characteristic, deep voids on the fracture surface, formed by the coalescence of micro-voids (V), clearly visible at their bottom. On the other hand, no significant participation of foreign phase inclusions in the fracture initiation process is observed. Therefore, it can be assumed that the fracture mechanism in the fracture in question takes place along specific cleavage planes, resulting from the granular or block micro-structure of the material. The sample zones with a fragmented micro-structure are characterized by the presence of transcrystalline ductile fracture (DF) with a characteristic structure and arrangement of voids in the form of overlapping “scales” (FS). This indicates that the fracture process was preceded by a large plastic deformation. Due to the relatively small share of plastic fracture on the analyzed fracture surface, it can be stated that the cleavage fracture mechanism had a decisive influence on the value of fracturing work of the welded joint in the delivered condition, subjected to impact tests at reduced temperature.

The fractures of the heat-treated samples, in the entire temperature range of the impact toughness test, in the fracture zone are characterized by a similar void structure (Figure 42b,d and Figure 43b,d). In both cases, it can be concluded that the surface of the samples is dominated by a plastic fracture with a fairly uniform arrangement of voids. The formation of the observed structure begins in the micro-voids (V), at the bottom of which the precipitation of foreign phases (I) is observed only sporadically. Due to faster fracture process, at the fracture of the sample tested at negative temperature (Figure 43c), the characteristic “fish scale” (FS) structure appears to a greater extent, running approximately in the direction of fracture. The parabolic outline of the scale structure indicates uneven loading during fracture formation, resulting from unequal tension or so-called tears. With regard to the sample tested at negative temperature, it is also possible to state the presence of a plastic fracture resulting from the action of shear forces [29]. This is evidenced by the presence of long strands of scale structure (FS), often ended (or separated) by large voids. The presence of micro-voids in the strands of the scale structure is observed, and the interface between the strands may be tear ridges (TR). However, such a situation is not observed in the fracture at ambient temperature (Figure 43b), in which the scale structure is observed to a much lesser extent. In the surface of this fracture, no large number of deep, topographically developed voids is also observed. Shallow, very small, locally parabolic voids dominate. It is also possible to indicate the local occurrence of relatively larger areas of the cleavage fracture (without clear steps) but separated by inter-crystalline cracks (IC). Most likely, it results from the morphology of the martensitic structure obtained in the course of post-welding heat treatment, which was characterized by a relatively low variation in the crystallographic orientation of the formed blocks within the former austenite grains. The described type of structure, in low-alloy steels, may be characterized by a lower level of impact toughness, especially in the case of increased carbon content, in relation to the martensitic structure of high variability [32].

Figure 44 and Figure 45 present images of the fracture surfaces of the tested samples, recorded in the lateral zones—the areas marked with “D” frames in Figure 37. Within the same series of samples, i.e., in the delivered condition and heat-treated, the fractures share similar morphological features. Immediately after welding, for both test temperatures, the fractures of samples can be described as transcrystalline cleavable, with very small areas of plastic fractures—Figure 44a,c and Figure 45a,c. In both cases, numerous steps (S) of varying heights are present. In areas formed by blocks of similar micro-structure, steps have a flat course, creating a branched system of “river pattern” (RP). In other areas, quite high steps—indicating the expenditure of considerable energy during the fracture process–largely resemble so-called tear ridges (TR). In addition, the presence of “tongues” is observed, which, taking into account the features described above, allows the fracture to be classified as quasi-cleavage. It is also worth adding that at the fracture in ambient temperature (Figure 44a and Figure 45a), in selected cleavage zones, the development of void fractures, characteristic of a plastic fracture, is sporadically observed. It is especially noticeable at lower magnification in the form of characteristic facets deformed at the ridges—Figure 44a. Additionally, in both impact test temperatures, the fractures show numerous extensive inter-crystalline cracks, the presence of which can be mainly explained by a very large diversity of micro-structure within the heat-affected zone. Observed structure of the samples’ fracture in the delivered condition proves that the formation of fractures took place well below the brittleness threshold of the material.

On the other hand, the fractures of the samples subjected to heat treatment (Figure 44b,d and Figure 45b,d) can be described as quasi-cleavable with very large areas of plastic fracture. This is evidenced by extensive ductile zones (DF), in which most of the characteristic void structure are observed, arranged in the form of “fish scales” (FS). Small, even voids (V) dominate, at the bottom of which there are very fine particles of inclusions (I).

The cleavage zones of the fractures in question, on the other hand, are characterized by high steps (S), places that are very difficult to distinguish from tear ridges (TR). In the fracture zones in question, the presence of “river” pattern is not observed. Few flat surfaces of these fractures locally show the presence of “tongues”, formed as a result of a twinning mechanism of micro-voids, located directly on the fracture front.

## 4. Conclusions

The aim of the research, widely discussed in this study, was to analyze the possibility of using welding techniques to connect high-strength, abrasion-resistant, martensitic Hardox 600 steel. In addition to very generally formulated aim of the research, an additional area under consideration was practical aspects of increasing selected strength parameters of a welded joint based on conventional—available in industrial conditions —comprehensive heat treatment operations. Based on the research, it was shown that, in the case of Hardox 600 steel, characterized by a high CEV carbon equivalent, it is possible to use welding techniques for its joining. It has been shown that, by appropriate selection of technology, additional materials and welding parameters, as well as comprehensive heat treatment procedures, it is possible to obtain a correct (devoid of incompatibility) welded joint, characterized by very high mechanical indexes corresponding to the base material. It is worth noting that the achievement of the assumed goal required, in addition to technological measures, implementation of a diverse test procedure, consisting of structural, chemical and strength tests, impact tests and extensive fractographic analysis. All the issues discussed in this manuscript can be summarized as follows:As a result of welding, in the entire area of the welded joint, a very diverse micro-structure is obtained. Overall, it leads to a reduction in both hardness levels and virtually all strength and impact indicators related to the base material properties. At the same time, this state of affairs allows us to conclude that the implementation of welding processes, in the case of Hardox 600 steel, leads to a reduction in its resistance to abrasive wear processes. The author’s experience in this area has shown that the above-described phenomenon takes place both in the area of the weld material and in a wide area of the base material, determined by the heat-affected zone.Comprehensive thermal treatments performed after welding, together with a welding technology appropriately selected for this purpose, allowed for very favorable transformation of the micro-structure. As a result of the technological operations, within the entire welded joint, structures similar to the structure of Hardox 600 steel were obtained in the delivery condition from the steelworks.From the point of view of mechanical and functional properties, the obtained average levels of impact toughness of the welded joint in the state immediately after welding, amounting to (at the temperatures of +20 °C and −40 °C): 25.0 J/cm^2^ and only 6.0 J/cm^2^, clearly indicate the brittleness threshold already at ambient temperature. Additionally, the obtained low strength indices of the welded joint (e.g., R_p0.2_ = 674 MPa, R_m_ = 879 MPa), from a practical point of view, exclusion of safe implementation of the Hardox 600 steel welding process, assuming no post-welding heat treatment. It is worth noting that the statement made, in addition to the above-mentioned numerical data, was also confirmed in the course of an extensive fractographic analysis.Research team allowed for the selection of welding technology and conditions and the development of an optimal set of thermal treatments for the performed welded joints of Hardox 600 steel, enabling reconstruction of structural, strength and impact properties, negatively transformed as a result of welding, to those corresponding to the base material. As a result of post-heat treatment, the following mechanical indicators were obtained, together with a percentage reference to the actual properties of the base material (summarized in Table 1 for the longitudinal direction of the sheet): R_p0.05_ = 1034 MPa (79%), R_p0.2_ = 1354 MPa (88%), R_m_ = 1800 MPa (85%), A_5_ = 7.8% (64%), Z = 21.4% (55%), KCV_+20_ = 40.0 J/cm^2^ (99%), KCV_−40_ = 32.0 J/cm^2^ (114%). Particularly with regard to the impact toughness indices obtained, it is worth mentioning that, after [27], the brittleness threshold of structural steels is assumed to be the impact toughness value of 35 J/cm^2^, which, at the same time, corresponds approximately to 50% of the ductile fracture. Taking into account the results of the fractographic analysis in individual zones of the samples subjected to impact tests, it can, therefore, be assumed that the proposed comprehensive heat treatment allowed us to obtain very favorable plastic properties of the welded joint of Hardox 600 steel, practically occurring above the brittleness threshold.

## Figures and Tables

**Figure 1 materials-14-04541-f001:**
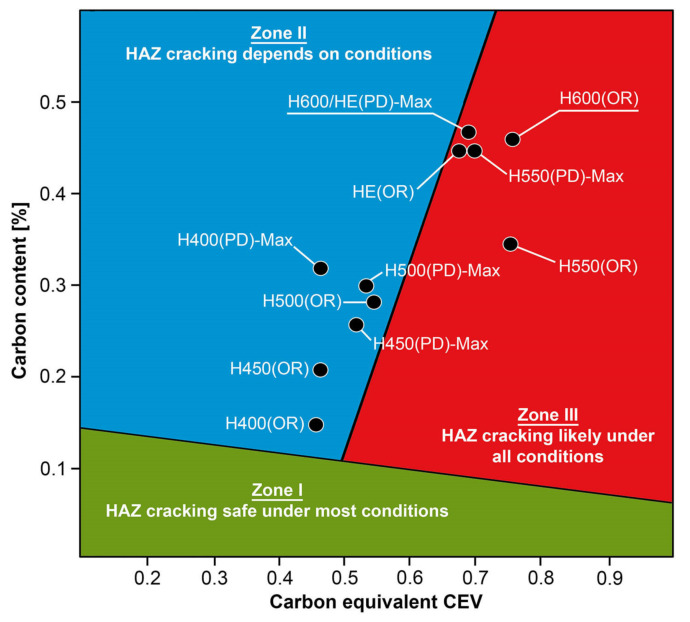
Tendency to crack as a function of carbon content and carbon equivalent CEV of welded joints of selected Hardox steels. PD—producer’s data for sheet thicknesses: 8.0–20.0 mm (Hardox 400), 10.0–19.9 mm (Hardox 450), 4.0–13.0 mm (Hardox 500), 8.0–31.9 mm (Hardox 550), 6.0–35.0 mm (Hardox 600), 8.0–19.0 mm (Hardox Extreme); OR—the author’s own tests for sheets with a thickness of 10.0 mm; Max—maximum value. The study is based on the data in Table 2 and refs [23,24].

**Figure 2 materials-14-04541-f002:**
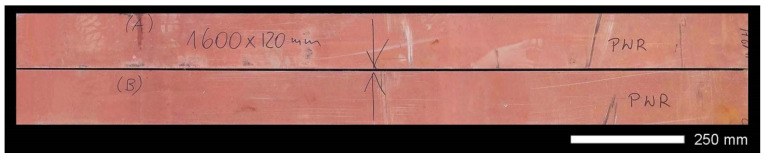
Pieces of Hardox 600 sheet for the welding process. The arrows indicate the place of the weld, (**A**)/(**B**)—pieces to be weld.

**Figure 3 materials-14-04541-f003:**
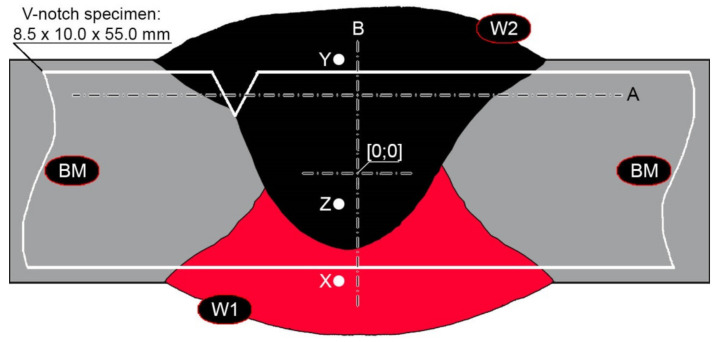
General diagram of the welded joints made of Hardox 600 steel. W1/W2—individual layers of the weld according to the sequence of their execution, A/B—lines of performed hardness distributions, BM—base material, X/Y/Z—locations of the chemical analyzes performed, the location of the “V” notch cut on the impact test specimens is shown schematically in white.

**Figure 4 materials-14-04541-f004:**
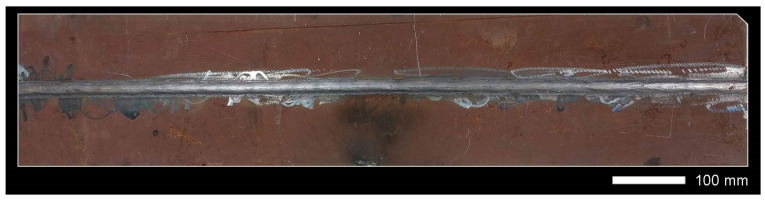
A fragment of steel sheets shown in Figure 2 with a welded joint.

**Figure 5 materials-14-04541-f005:**
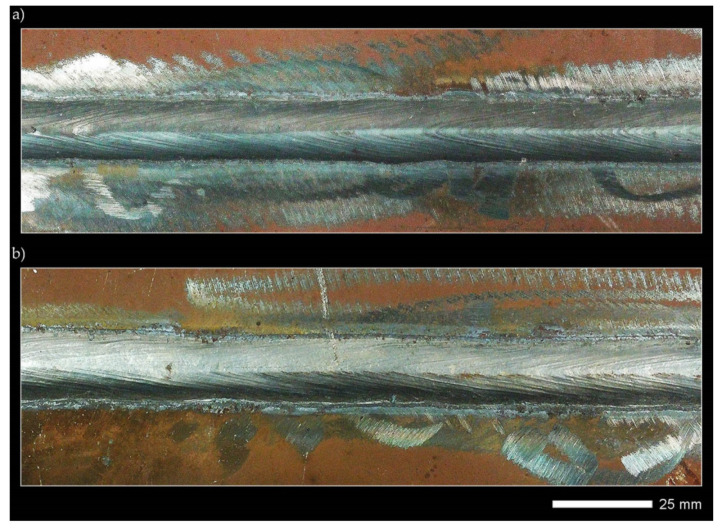
Macroscopic image of a fragment of the welded joint shown in Figure 4. (**a**) View from the front of weld No. 1—made first. (**b**) View from the front of weld No. 2.

**Figure 6 materials-14-04541-f006:**
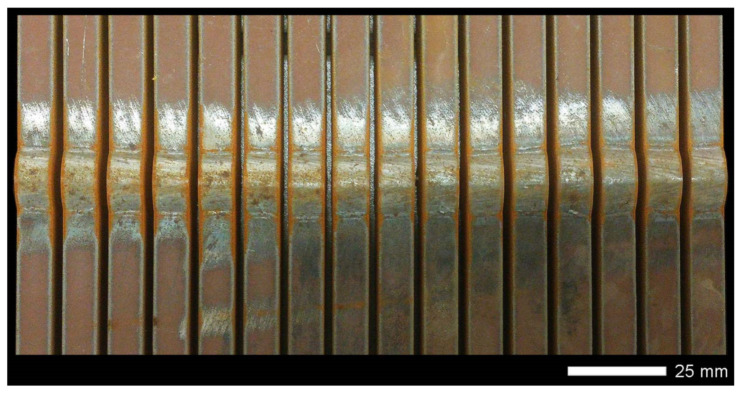
A fragment of a welded joint of Hardox 600 steel after the operation of cutting rectangular specimens from the weld area using a high-energy liquid stream with abrasive.

**Figure 7 materials-14-04541-f007:**
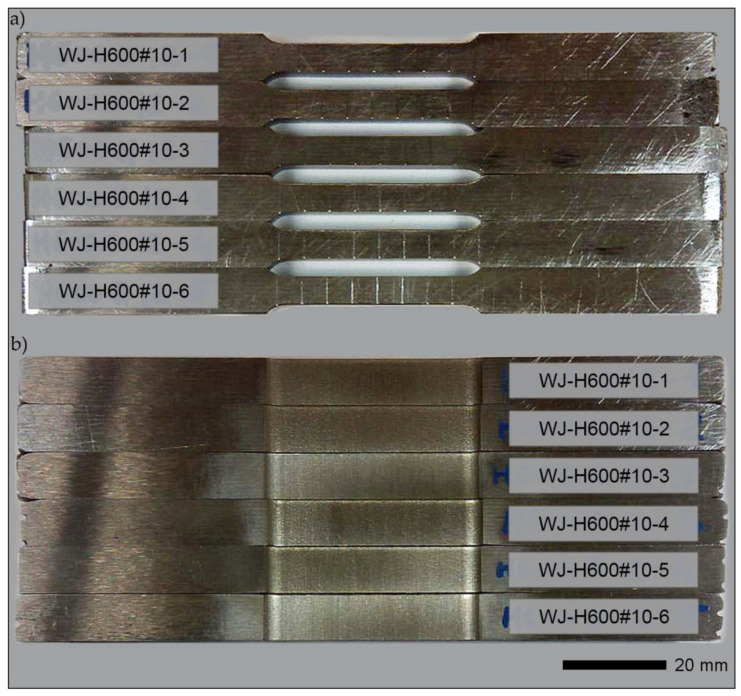
Examples of strength samples shown in Figure 6, subjected to a finishing operation. In the heat-affected zone, undercuts were made, constituting a measurement base of the assumed length. The cutting was performed with the electro-erosion method. (**a**) View of samples from the side of the sheet surface (front weld) with a scale to determine a relative elongation. (**b**) View of the samples from the side of the sheet cross-section.

**Figure 8 materials-14-04541-f008:**
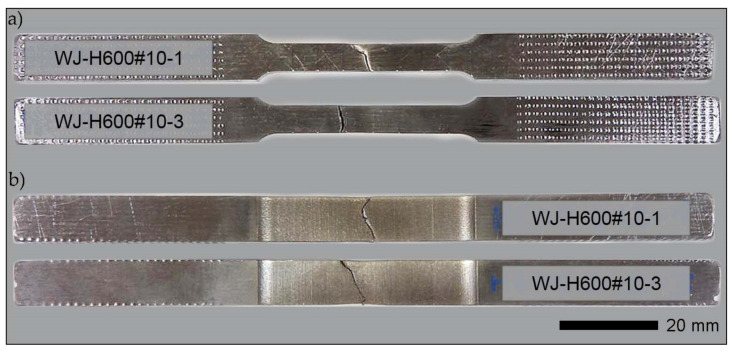
Samples subjected to a static tensile test: (**a**) View from the side of the sheet surface (weld front). (**b**) View from the side of the sheet cross-section.

**Figure 9 materials-14-04541-f009:**
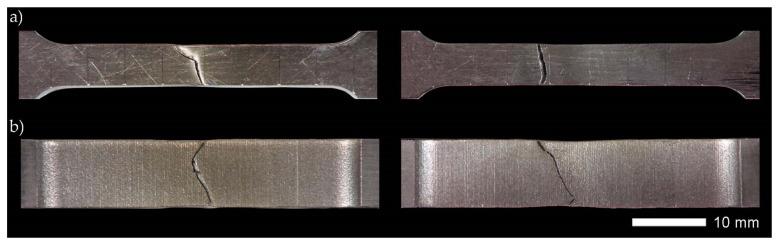
Enlarged macroscopic images of the databases of the samples shown in Figure 8: (**a**) View of the sample measurement base from the side of the sheet surface (weld front). (**b**) Sample measurement base view from the sheet cross-section side.

**Figure 10 materials-14-04541-f010:**
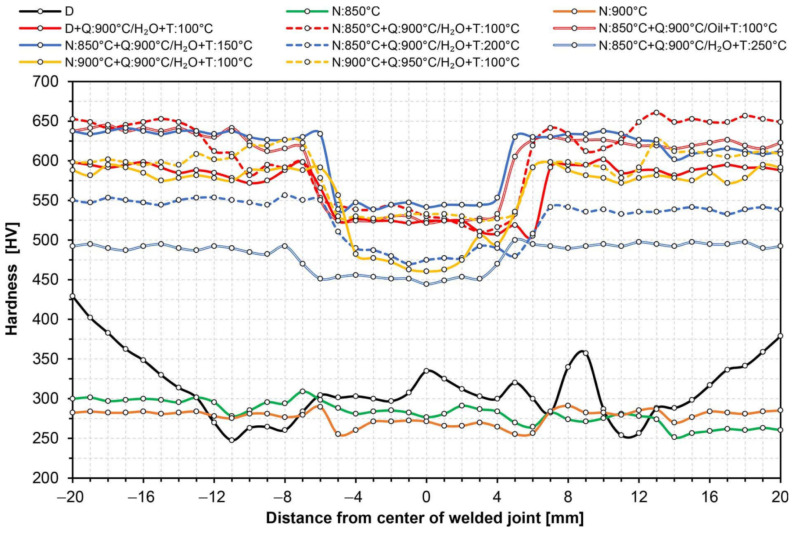
Distribution of the hardness of the welded joint Hardox 600 along the line A marked in Figure 3, subjected to various heat treatments, summarized in Table 6. D—delivery condition (without heat treatment), N—normalization, Q—quenching, T—tempering.

**Figure 11 materials-14-04541-f011:**
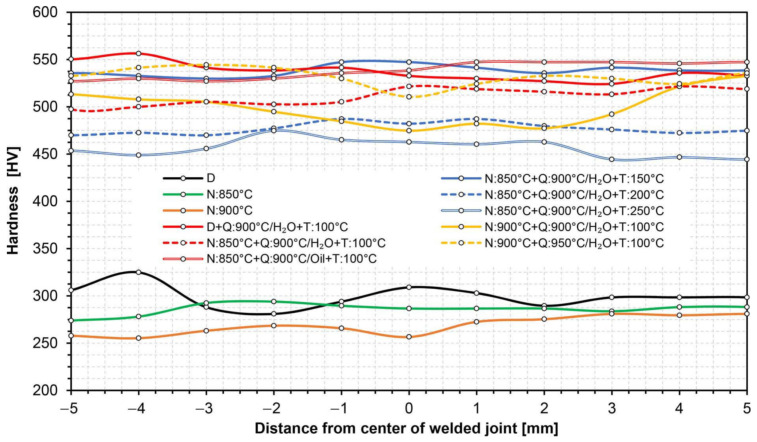
Distribution of the hardness of the welded joint Hardox 600 along the line B marked in Figure 3, subjected to various heat treatments, summarized in Table 6. D—delivery condition (without heat treatment), N—normalization, Q—quenching, T—tempering.

**Figure 12 materials-14-04541-f012:**
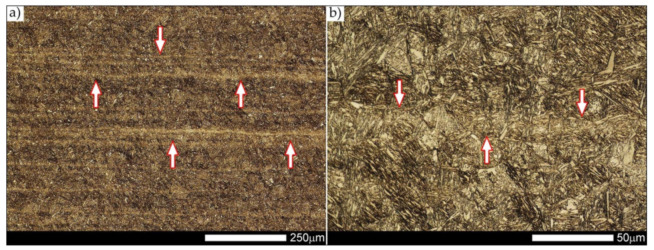
Micro-structure of Hardox 600 as delivered from a steelworks (before welding)—longitudinal direction for plastic working [4]. (**a**) A general view of the micro-structure with banding features clearly visible. (**b**) Enlarged fragment of the central zone shown in (**a**). Structure of tempering martensite with bands of quenching martensite. The arrows indicate the strands of unremoved martensite running in the direction of thermomechanical rolling. Light microscopy, etched with 3% HNO_3_ in ethyl alcohol.

**Figure 13 materials-14-04541-f013:**
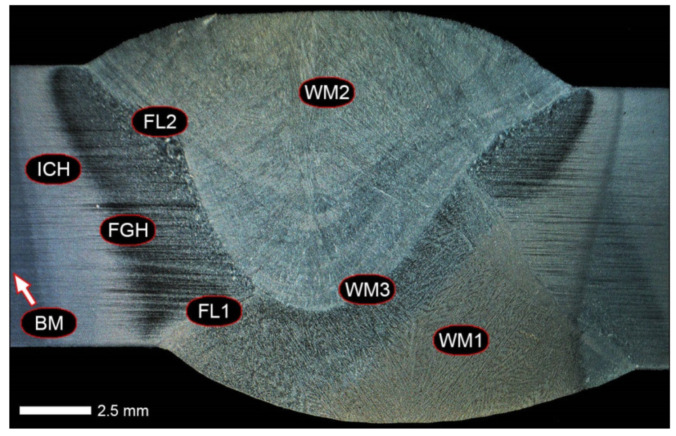
Macroscopic view of the weld joint cross section of Hardox 600 as delivered. WM—weld metal zone, FL—fusion line, BM—base material zone, FGH(AZ)—fine-grained zone (normalization and re-crystallization), ICH(AZ)—inter-critical zone (incomplete normalization). Stereoscopic microscopy, etched with 3% HNO_3_ in ethyl alcohol and Adler’s reagent.

**Figure 14 materials-14-04541-f014:**
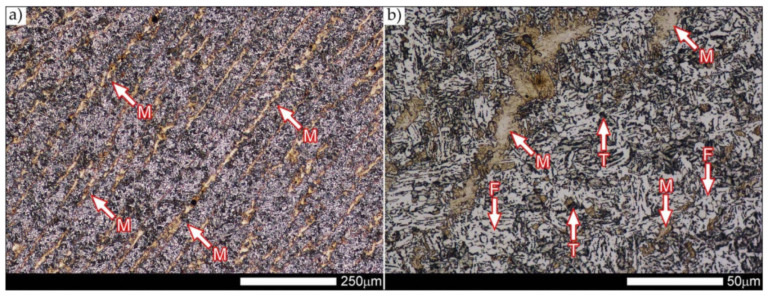
Micro-structure of a welded joint of Hardox 600 as delivered in the area marked WM1 in Figure 13: (**a**) General view of the micro-structure with clear banding features. (**b**) Enlarged central area of the structure shown in (**a**). A very diverse structure, mainly consisting of banded martensite against the background of non-equilibrium ferrite. The presence of troostite colonies is also locally observed. M—martensite, F—ferrite, T—troostite. Light microscopy, etched with 3% HNO_3_ in ethyl alcohol.

**Figure 15 materials-14-04541-f015:**
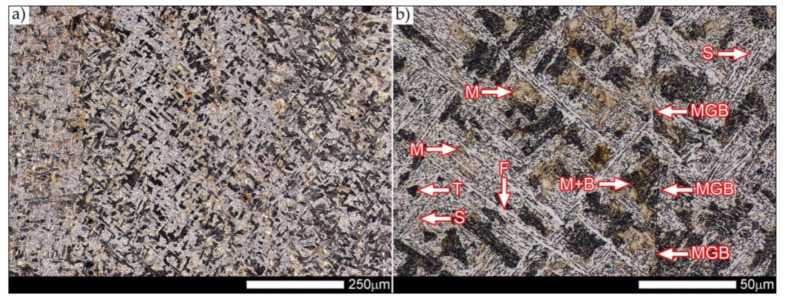
Micro-structure of the welded joint of Hardox 600 as delivered in the area marked as WM2 in Figure 13: (**a**) General view of a dendritic structure of a morphologically very diverse structure. (**b**) Enlarged central area of the structure shown in (**a**). Micro-structure consisting of martensite and lower bainite and a few colonies of troostite against the background of quenching sorbite. M—martensite, F—ferrite, T—troostite, S—sorbite, B—lower bainite, MGB—migrated grain boundary. Light microscopy, etched with 3% HNO_3_ in ethyl alcohol.

**Figure 16 materials-14-04541-f016:**
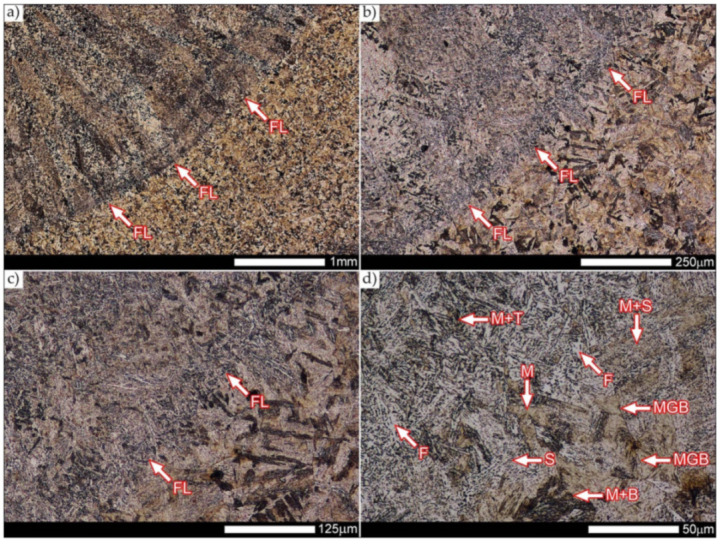
Micro-structure of a welded joint of Hardox 600 as delivered in the area marked WM3 in Figure 13—different magnification scales: (**a**) ~25×; (**b**) ~100×; (**c**) ~200×; (**d**) ~500×. A clearly visible fusion line with a very diverse micro-structure. The quenching sorbite structure is observed along the entire length of the fusion line, as is the sorbitic–martensitic and troostitic–martensitic structure. In the zones directly adjacent to the fusion line, there are areas of quenching martensite and lower bainite, locally separated by migrated boundaries. FL—fusion line, M—martensite, F—ferrite, T—troostite, S—sorbite, B—lower bainite, MGB—migrated grain boundary. Light microscopy, etched with 3% HNO_3_ in ethyl alcohol.

**Figure 17 materials-14-04541-f017:**
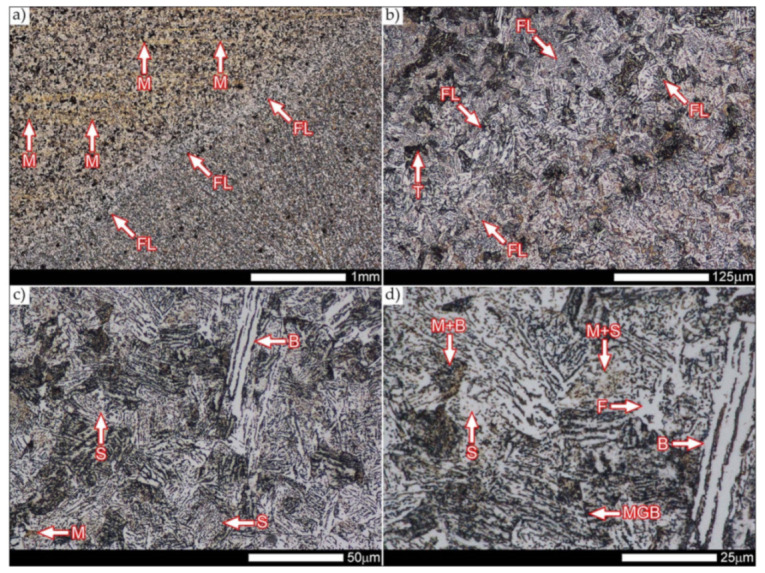
Micro-structure of a welded joint of Hardox 600 as delivered in the area marked as FL1 in Figure 13—different magnification scales: (**a**) ~25×; (**b**) ~200×; (**c**) ~500×; (**d**) ~1000×. A clearly visible fusion line (FL) with a very diverse micro-structure. Quenching sorbite (S), martensitic–sorbitic (M+S) structure, areas of quenching martensite and lower bainite (M+B) are observed along the entire length of the fusion line. Locally, the formation of structurally free quenching martensite (M) and quenching troostite (T) as well as upper bainite (B) are observed. There are also migrated grain boundaries (MGB). Light microscopy, etched with 3% HNO_3_ in ethyl alcohol.

**Figure 18 materials-14-04541-f018:**
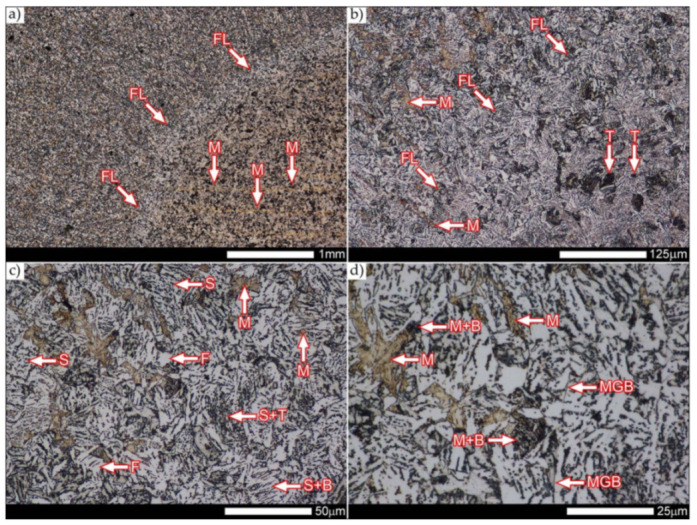
Micro-structure of a welded joint of Hardox 600 as delivered in the area marked FL2 in Figure 13—different magnification scales: (**a**) ~25×; (**b**) ~200×; (**c**) ~500×; (**d**) ~1000×. A clearly visible fusion line (FL) with a very diverse micro-structure. Along the entire length of the fusion line and in the zones adjacent to it, the structure of quenching sorbite (S), sorbitic–bainitic (S+B), sorbitic–troostitic (S+T), areas of quenching martensite and lower bainite (M+B) banded—structurally free—quenching martensite (M) are observed. The secretions of quenching troostite (T) as well as block ferrite (F) are locally observed. MGB—migrated grain boundary. Light microscopy, etched with 3% HNO3 in ethyl alcohol.

**Figure 19 materials-14-04541-f019:**
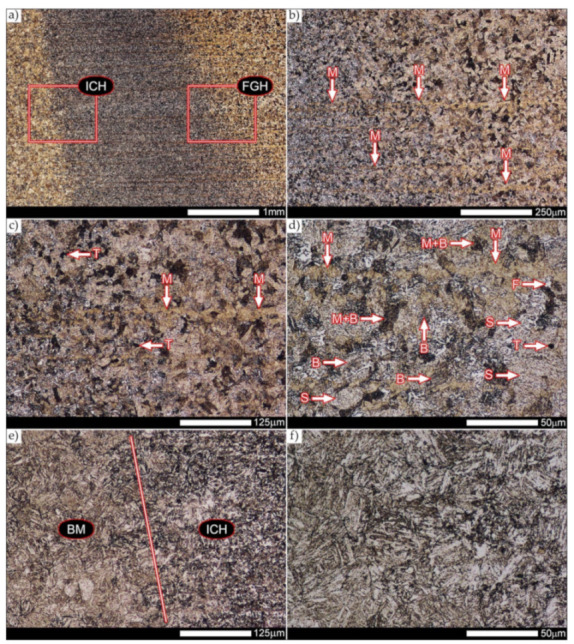
Micro-structure of the welded joint of Hardox 600 steel in the delivered condition in the area of the heat-affected zone, marked as FGH and ICH in Figure 13. (**a**) General view of the heat-affected zone. (**b**–**d**) Enlarged areas of the micro-structure marked by frame FGH in (**a**). (**e**,**f**) Enlarged areas of the micro-structure marked with frame ICH in (**a**). In the whole area of the zone of warm influence, the presence of martensitic structure bands (M) is observed. The following structures can be distinguished in the FGH zone: quenching martensite with lower bainite (M+B), upper bainite (B), quenching sorbite (S), quenching troostite (T) and block ferrite (F). The ICH zone consists mainly of a post-martensitic structure, i.e., tempered sorbite (ICH in (**e**)) and tempered martensite (BM in (**e**)). Light microscopy, etched with 3% HNO_3_ in ethyl alcohol.

**Figure 20 materials-14-04541-f020:**
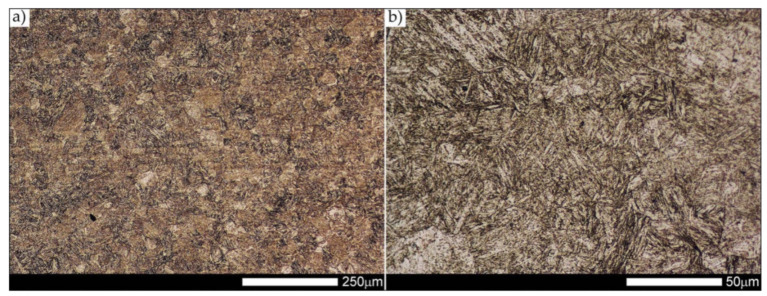
Micro-structure of the base material of the weld joint of Hardox 600 in the as welded condition—area BM in Figure 13. (**a**) General view of the micro-structure with very little banding features. (**b**) Enlarged central area of the structure shown in (**a**). In terms of a structure, the micro-structure is similar to that of fine-lath tempering martensite. Light microscopy, etched with 3% HNO_3_ in ethyl alcohol.

**Figure 21 materials-14-04541-f021:**
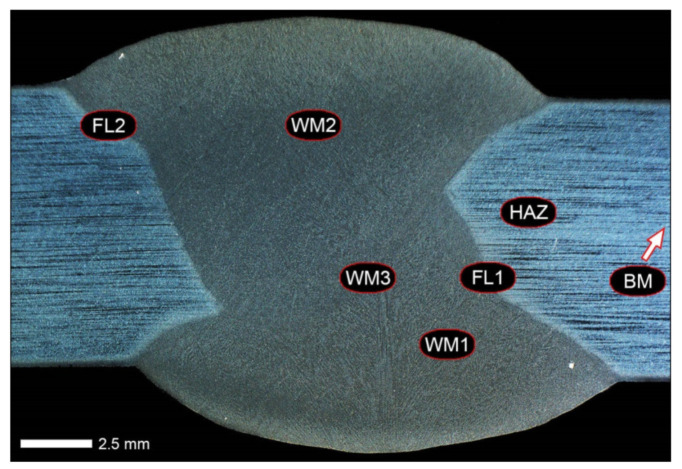
Macroscopic view of the welded joint cross section of Hardox 600 in normalized state—process No. 3 shown in Table 6. WM—weld metal zone, FL—fusion line, BM—base material zone, HAZ—heat-affected zone. Stereoscopic microscopy, etched with 3% HNO_3_ in ethyl alcohol.

**Figure 22 materials-14-04541-f022:**
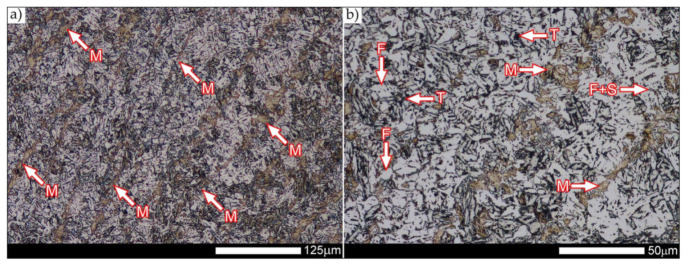
Micro-structure of the welded joint of Hardox 600 in normalized state in the area marked as WM1 in Figure 21. (**a**) General view of the micro-structure with clear banding features. (**b**) Enlarged central area of the structure shown in (**a**). Structure diversified in terms of structure characteristic for incompletely hardened state. Banded development of martensite against the background of non-equilibrium ferrite and sorbite are observed. The presence of quenching troostite colonies is also locally observed. M—martensite, F—ferrite, T—troostite, S—sorbite. Light microscopy, etched with 3% HNO_3_ in ethyl alcohol.

**Figure 23 materials-14-04541-f023:**
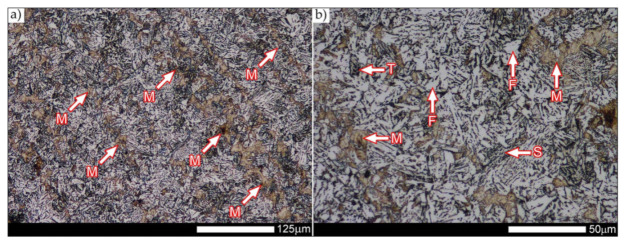
Micro-structure of the welded joint of Hardox 600 in normalized state in the area marked as WM2 in Figure 21. (**a**) General view of the micro-structure with distinct banding features. (**b**) Enlarged central area of the structure shown in (**a**). Differentiated structure characteristic for incompletely hardened state. Band-like development of martensite is observed—in the direction of the weld crystallization—against the background of non-equilibrium ferrite and quenching sorbite. In terms of structure, the ferritic phase is very similar to the upper bainite. The presence of quenching troostite colonies is also locally observed. M—martensite, F—ferrite, T—troostite, S—sorbite. Light microscopy, etched with 3% HNO_3_ in ethyl alcohol.

**Figure 24 materials-14-04541-f024:**
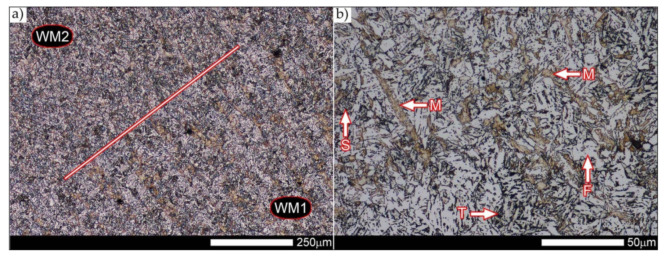
Micro-structure of the welded joint of Hardox 600 steel in the normalized state in the area marked as WM3 in Figure 21. (**a**) General view of the micro-structure—the line shows a slightly outlined fusion line. (**b**) The enlarged area of the structure on the fusion line shown in (**a**). Relatively homogeneous, phase-different structure characteristic of an incompletely hardened state. The presence of non-equilibrium ferrite grains is observed—for the most part—locally with the features of upper bainite, with quenching sorbite and with isolation martensite precipitates and in the form of bands running in the direction of crystallization. The presence of quenching troostite colonies can also be observed in a small amount. WM1/WM2—weld material No.1 and No.2, M—martensite, F—ferrite, T—troostite, S—sorbite. Light microscopy, etched with 3% HNO_3_ in ethyl alcohol.

**Figure 25 materials-14-04541-f025:**
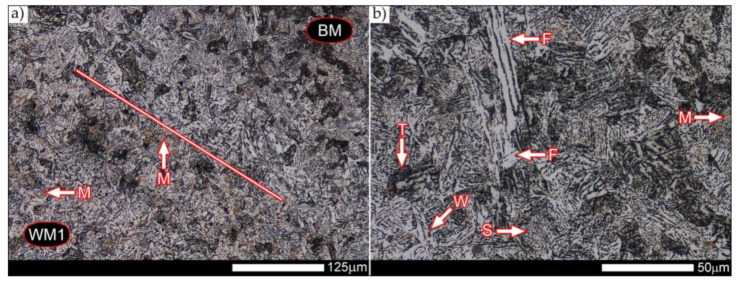
Micro-structure of the welded joint of Hardox 600 steel in a normalized state in the area marked as FL1 in Figure 21. (**a**) General view of the micro-structure—the line shows a slightly indicated fusion line. (**b**) Enlarged area of the structure on the fusion line shown in (**a**). In terms of construction, the structure is characteristic for isothermal quenching—quenching sorbite (S) with the features of the upper bainite. In addition, the presence of non-crystallized ferrite (F), significant amounts of quenching troostite colonies (T), martensite (M) locally transforming into lower bainite and individual areas with Widmanstätten (W) structure features can be indicated. WM1—weld material No.1, BM—base material. Light microscopy, etched with 3% HNO_3_ in ethyl alcohol.

**Figure 26 materials-14-04541-f026:**
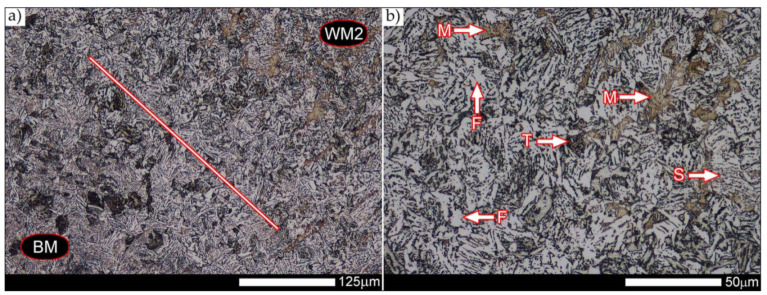
Micro-structure of the welded joint of Hardox 600 steel in the normalized state in the area marked as FL2 in Figure 21. (**a**) General view of the micro-structure—the line shows a slightly outlined fusion line. (**b**) The enlarged area of the structure on the fusion line shown in (**a**). Relatively homogeneous, differentiated in terms of phase structure, the structure characteristic for the incompletely hardened state. Against the background of non-equilibrium ferrite grains (F) and quenching sorbite (S), visible bands of the martensitic structure (M) and troostite colonies (T). WM2—weld material No. 2, BM—base material. Light microscopy, etched with 3% HNO_3_ in ethyl alcohol.

**Figure 27 materials-14-04541-f027:**
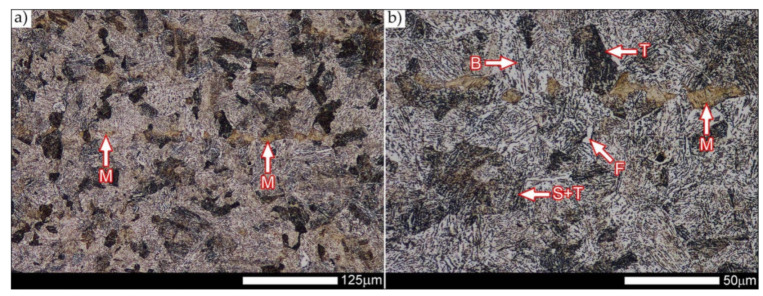
Micro-structure of a welded joint of Hardox 600 in the normalized state in the area marked as HAZ in Figure 21. (**a**) General view of the micro-structure with marked banding features. (**b**) The enlarged central area of the structure shown in (**a**). The structure characteristic for isothermal cooling—on the background of quenching sorbite structure (S) and quenching troostite (T), band-like precipitation of martensite (M), locally changing into the structure of upper bainite (B). In a few zones, the presence of structurally free ferrite (F) is also observed. Light microscopy, etched with 3% HNO3 in ethyl alcohol.

**Figure 28 materials-14-04541-f028:**
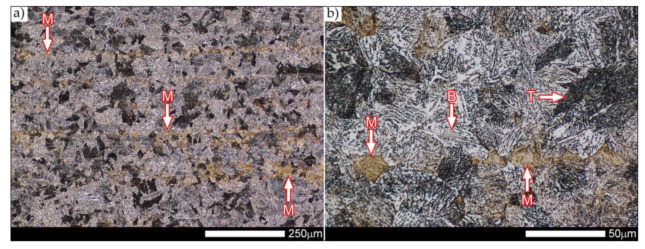
Micro-structure of the base material of the welded joint of Hardox 600 steel in normalized state—area BM in Figure 21. (**a**) General view of the micro-structure with streaked areas of quenching martensite. (**b**) Enlarged central area of the structure shown in (**a**). In terms of structure, the structure is characteristic of isothermal cooling—on the background of quenching sorbite and quenching troostite visible precipitation of banded martensite. Upper bainite crystallizing from the grain boundaries of the former austenite is locally observed. M—martensite, B—bainite, T—troostite. Light microscopy, etched with 3% HNO_3_ in ethyl alcohol.

**Figure 29 materials-14-04541-f029:**
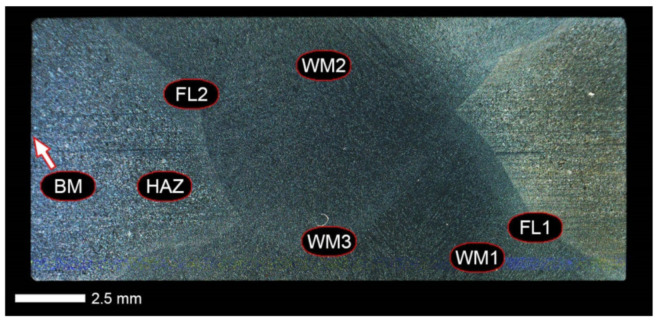
Macroscopic image of the welded joint cross-section of Hardox 600 in the heat-treated state according to the diagram No. 9 shown in Table 6. WM—weld metal zone, FL—fusion line, BM—base material zone, HAZ—heat-affected zone. Stereoscopic microscopy, etched with 3% HNO_3_ in ethyl alcohol.

**Figure 30 materials-14-04541-f030:**
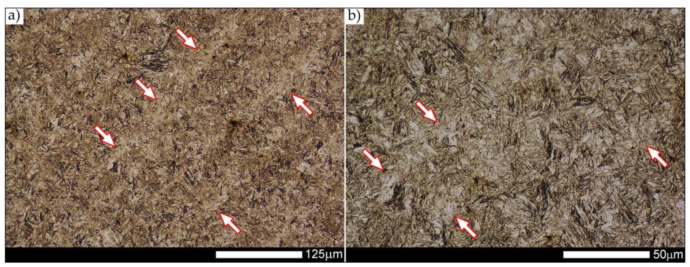
Micro-structure of the welded joint of Hardox 600 in the heat-treated condition according to the diagram No. 9 shown in Table 6, in the area labeled WM1 in Figure 29. (**a**) General view of the micro-structure with slight banding features. (**b**) Enlarged central area of the structure shown in (**a**). The structure of fine-lathed tempering martensite with quenching martensite separated along the strips (in the direction of the weld crystallization) (undermined martensite—marked with arrows). Light microscopy, etched with 3% HNO_3_ in ethyl alcohol.

**Figure 31 materials-14-04541-f031:**
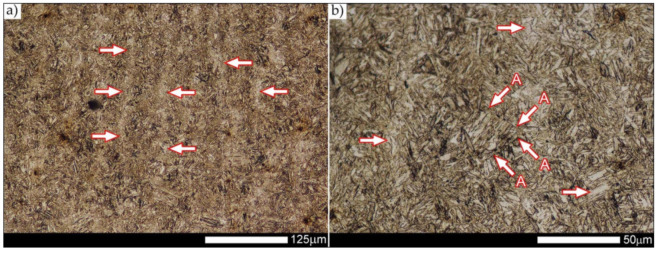
Micro-structure of a welded joint of Hardox 600 in the heat-treated condition as shown in Diagram No. 9 shown in Table 6 in the area labeled WM2 in Figure 29. (**a**) General view of the micro-structure with slight banding features. (**b**) Enlarged central area of the structure shown in (**a**). Structure of fine-lath tempering martensite with banded quenching martensite (indicated by arrows). Arrows (A) mark the grain boundary of the former austenite, inside which martensite arranged in the form of packages is observed, which is also characterized by a low degree of the crystallographic orientation’s variability. Light microscopy, etched with 3% HNO_3_ in ethyl alcohol.

**Figure 32 materials-14-04541-f032:**
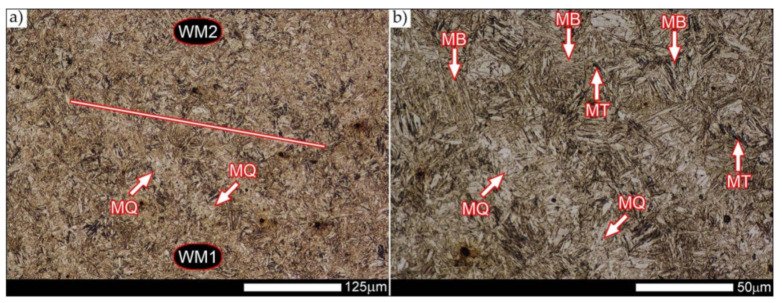
Micro-structure of the welded joint of Hardox 600 steel in the heat-treated state according to the diagram No. 9 shown in Table 6, in the area marked as WM3 in Figure 29. (**a**) General view of the micro-structure—the line symbolically marks the fusion line. (**b**) Enlarged central area of the structure at the fusion line shown in (**a**). For the most part, a structure consisting of fine-lathed tempering martensite with banded quenching martensite (MQ). Along the entire length of the fusion line, martensite of block structure (MB) is present, characterized by a small variation in the crystallographic orientation within the former austenite grains. The arrows (MT) indicate single precipitations of highly tempered martensite, which crystallized first. WM1/WM2—weld material No. 1 and No. 2. Light microscopy, etched with 3% HNO_3_ in ethyl alcohol.

**Figure 33 materials-14-04541-f033:**
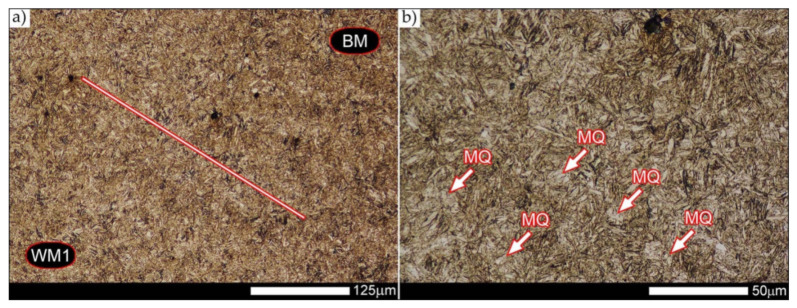
Micro-structure of the welded joint of Hardox 600 in the heat-treated condition according to the diagram No. 9 shown in Table 6, in the area marked as FL1 in Figure 29. (**a**) General view of the micro-structure—the fusion line is symbolically marked with a straight line. (**b**) Enlarged central area of the structure at the fusion line shown in (**a**). Throughout the fusion line, the structure, for the most part, consists of tempering martensite with very little banding characteristics formed by undemanding martensite (MQ). WM1—weld material No. 1, BM—base material. Light microscopy, etched with 3% HNO_3_ in ethyl alcohol.

**Figure 34 materials-14-04541-f034:**
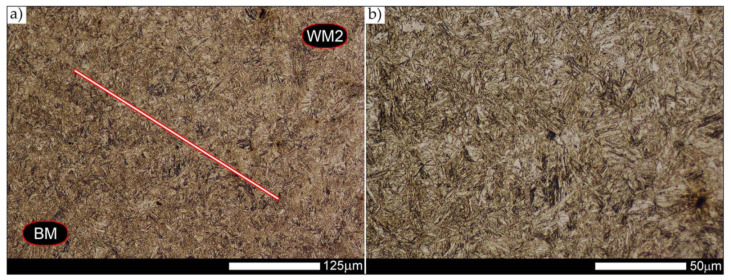
Micro-structure of the welded joint of Hardox 600 steel in the heat-treated condition according to the diagram No. 9 shown in Table 6, in the area marked as FL2 in Figure 29. (**a**) General view of the micro-structure—the fusion line is symbolically marked with a straight line. (**b**) Enlarged central area of the structure at the fusion line shown in (**a**). Throughout the fusion line, the structure predominantly consists of tempering martensite with very few areas of quenching martensite. There are no clear banding features in the structure. WM2—weld material No. 2, BM—base material. Light microscopy, etched with 3% HNO_3_ in ethyl alcohol.

**Figure 35 materials-14-04541-f035:**
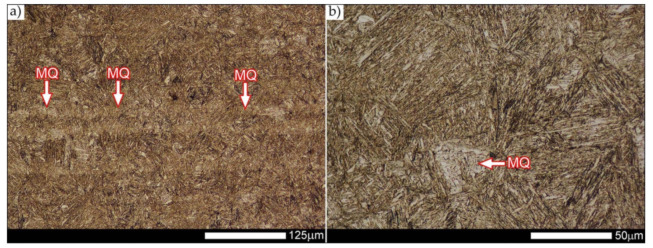
Micro-structure of the welded joint of Hardox 600 in the heat-treated state according to the diagram No. 9 shown in Table 6, in the area marked as HAZ in Figure 29. (**a**) General view of the micro-structure—the arrows indicate small banding features of the structure, created by the presence of undetermined martensite (quenching martensite—MQ). (**b**) Enlarged central area of the structure shown in (**a**). Structure mainly consists of fine-lathed tempering martensite. Light microscopy, etched with 3% HNO_3_ in ethyl alcohol.

**Figure 36 materials-14-04541-f036:**
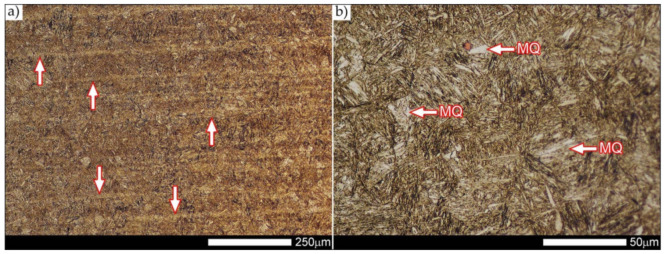
Micro-structure of the base material of the welded Hardox 600 steel in the heat-treated state according to the diagram No. 9 shown in Table 6—the BM area in Figure 29. (**a**) General view of the micro-structure—arrows indicate clear banding features created by underexposed martensite (quenching martensite)—MQ). (**b**) The enlarged central area of the structure shown in (**a**). The micro-structure of fine-lathed tempering martensite, characterized by a large variation of the crystallographic orientation within the block structure it creates. Light microscopy, etched with 3% HNO_3_ in ethyl alcohol.

**Figure 37 materials-14-04541-f037:**
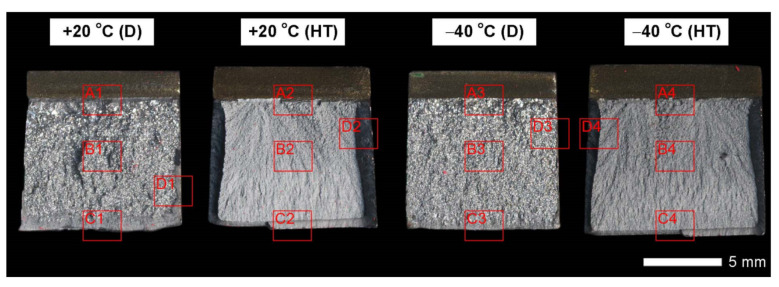
Macroscopic image of the fracture surfaces of representative samples of welded joints made of Hardox 600 after the impact test. D—delivery condition (after welding), HT—condition after comprehensive heat treatment according to diagram No. 9 shown in Table 6. The frames indicate: A—under-notch zone, B—central zone, C – final-fracture zone, D—side zone. Stereoscopic microscopy, non-etching condition.

**Figure 38 materials-14-04541-f038:**
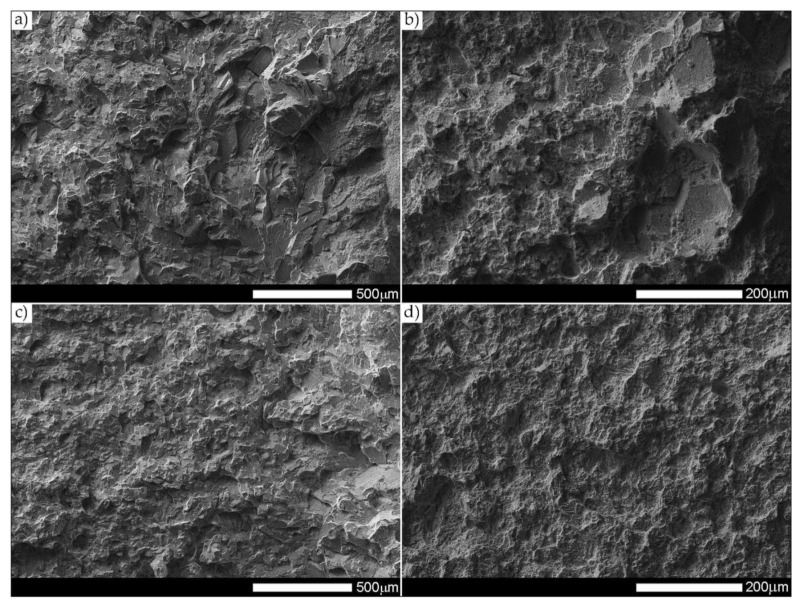
Images of the fracture surfaces of the welded joint Hardox 600, marked with the appropriate frames “A” in Figure 37: (**a**) A1, delivery condition, +20 °C, ~50×; (**b**) A2, heat-treated condition, +20 °C, ~150×; (**c**) A3, delivered condition, −40 °C, ~50×; (**d**) A4, heat-treated condition, −40 °C, ~150×. SEM, non-etching condition.

**Figure 39 materials-14-04541-f039:**
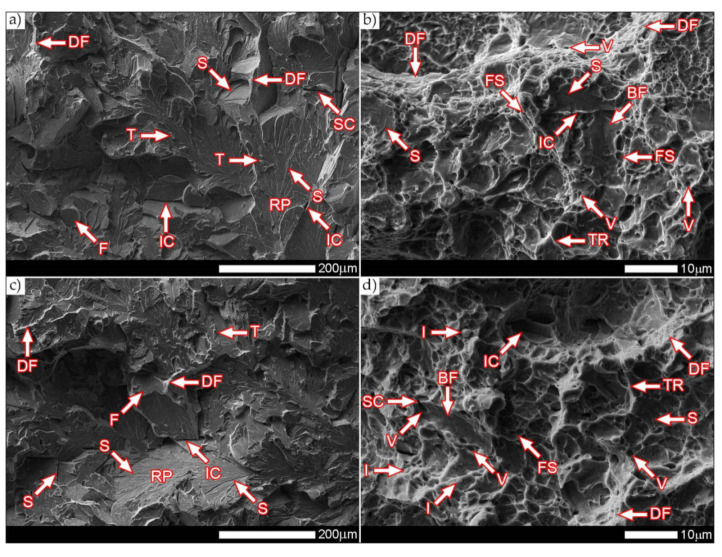
Enlarged images of the fracture surfaces shown in Figure 38: (**a**) A1, delivered condition, +20 °C, ~150×; (**b**) A2, heat-treated condition, +20 °C, ~1500×; (**c**) A3, delivered condition, −40 °C, ~150×; (**d**) A4, heat-treated condition, −40 °C, ~1500×. DF—ductile fracture, BF—brittle fracture, SC—secondary fracture, IC—inter-crystalline fracture, RP—“river” pattern, S—steps, T—tongues, F—facets, V—micro-voids, FS—fish scales, I—inclusions, TR—tear ridges. SEM, non-etching condition.

**Figure 40 materials-14-04541-f040:**
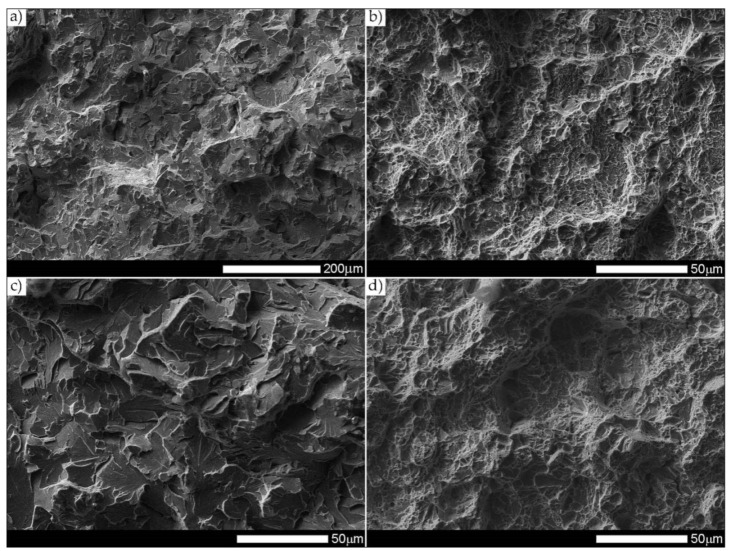
Images of the fracture surfaces of the welded joint Hardox 600, marked with the appropriate frames “B” in Figure 37: (**a**) B1, delivery condition, +20 °C, ~150×; (**b**) B2, heat-treated condition, +20 °C, ~500×; (**c**) B3, delivered condition, −40 °C, ~500×; (**d**) B4, heat-treated condition, −40 °C, ~500×. SEM, non-etching condition.

**Figure 41 materials-14-04541-f041:**
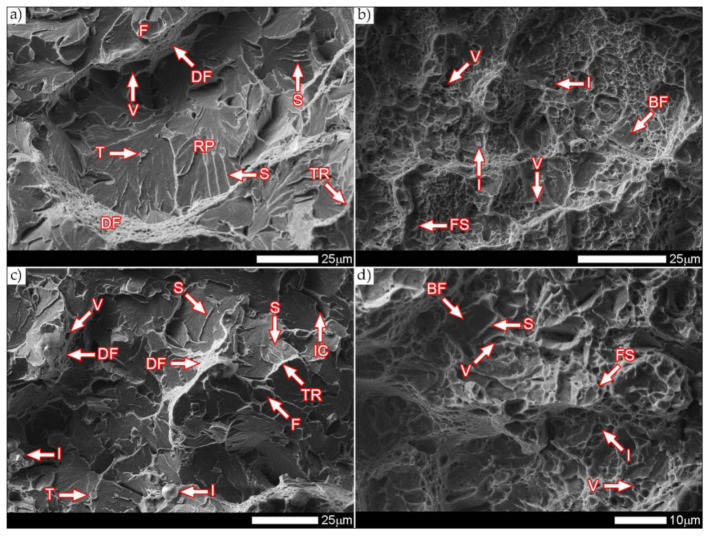
Enlarged images of the fracture surfaces shown in Figure 40: (**a**) B1, delivery condition, +20 °C, ~700×; (**b**) B2, heat-treated condition, +20 °C, ~1000×; (**c**) B3, delivered condition, −40 °C, ~750×; (**d**) B4, heat-treated condition, −40 °C, ~1500×. DF—ductile fracture, BF—brittle fracture, IC—inter-crystalline fracture, RP—“river” pattern, S—steps, T—tongues, F—facets, V—micro-voids, FS—fish scales, I—inclusions, TR—tear ridges. SEM, non-etching condition.

**Figure 42 materials-14-04541-f042:**
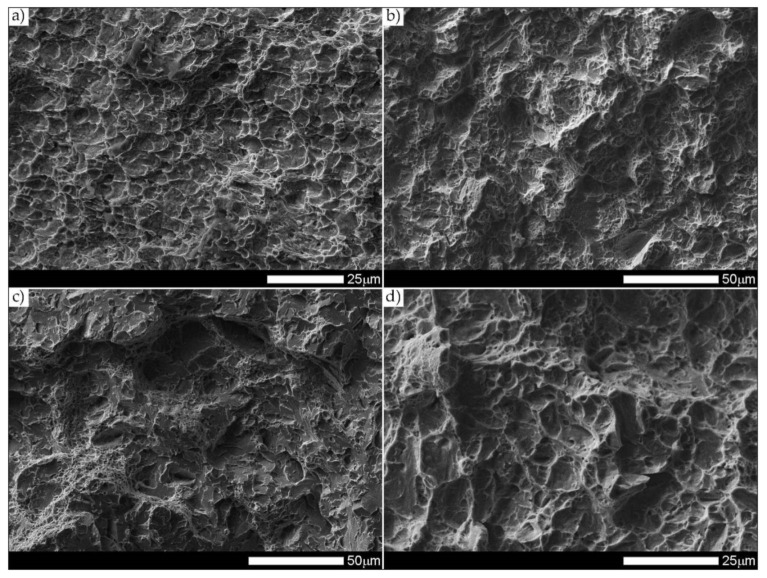
Images of the fracture surfaces of the welded joint Hardox 600, marked with the appropriate “C” frames in Figure 37: (**a**) C1, delivery condition, +20 °C, ~800×; (**b**) C2, heat-treated condition, +20 °C, ~500×; (**c**) C3, delivered condition, −40 °C, ~500×; (**d**) C4, heat-treated condition, −40 °C, ~1000×. SEM, non-etching condition.

**Figure 43 materials-14-04541-f043:**
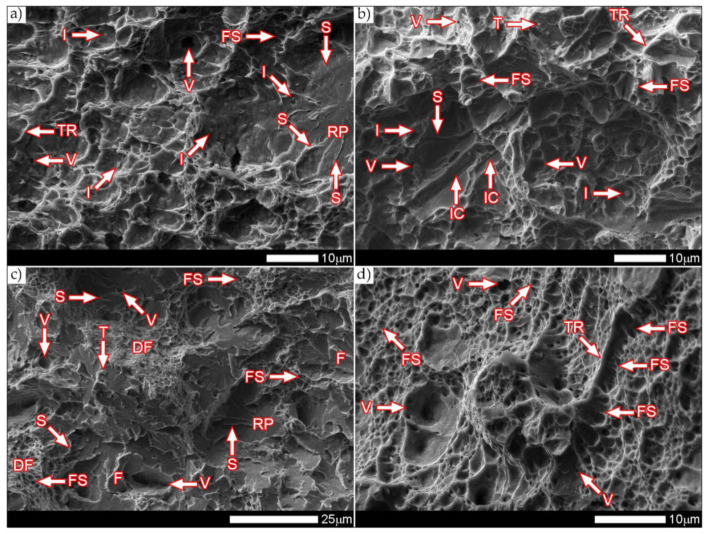
Enlarged images of the fracture surfaces shown in Figure 42: (**a**) C1, delivery condition, +20 °C, ~1500×; (**b**) C2, heat-treated condition, +20 °C, ~1500×; (**c**) C3, delivered condition, −40 °C, ~1000×; (**d**) C4, heat-treated condition, −40 °C, ~2000×. DF—ductile fracture, IC—inter-crystalline fracture, RP—“river” pattern, S—steps, T—tongues, F—facets, V—micro-voids, FS—fish scales, I—inclusions, TR—tear ridges. SEM, non-etching state.

**Figure 44 materials-14-04541-f044:**
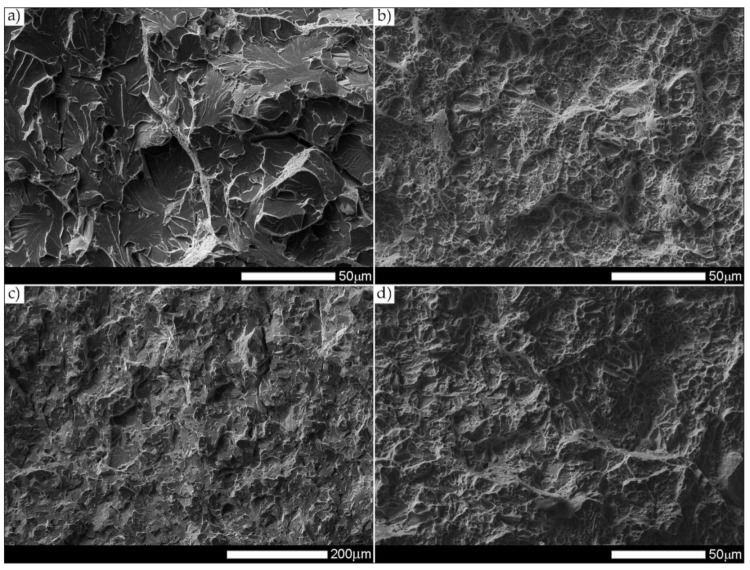
Images of the fracture surfaces of the welded joint Hardox 600, marked with the appropriate “D” frames in Figure 37: (**a**) D1, delivered condition, +20 °C, ~500×; (**b**) D2, heat-treated condition, +20 °C, ~500×; (**c**) D3, delivered condition, −40 °C, ~150×; (**d**) D4, heat-treated condition, −40 °C, ~500×. SEM, non-etching condition.

**Figure 45 materials-14-04541-f045:**
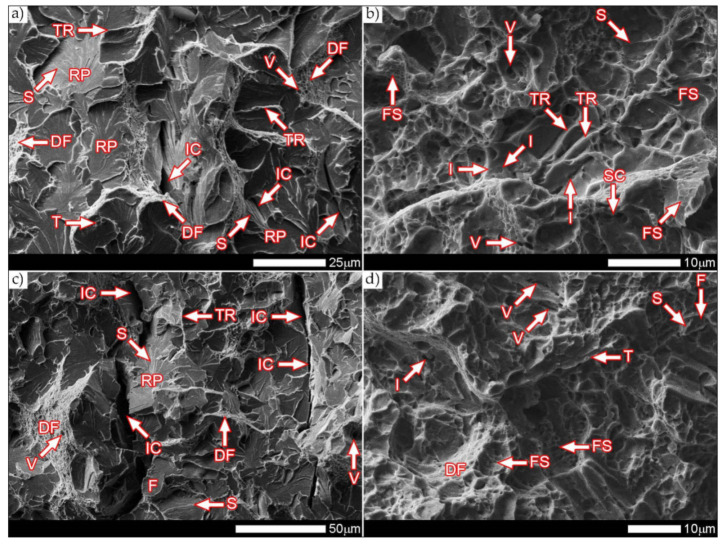
Enlarged images of the fracture surfaces shown in Figure 44: (**a**) D1, delivered condition, +20 °C, ~800×; (**b**) D2, heat-treated condition, +20 °C, ~2000×; (**c**) D3, delivered condition, −40 °C, ~500×; (**d**) D4, heat-treated condition, −40 °C, ~1500×. DF—ductile fracture, IC—inter-crystalline fracture, RP—“river” pattern, S—steps, T—tongues, F—facets, V—micro-voids, FS—fish scales, I—inclusions, TR—tear ridges. SEM, non-etching state.

**Table 1 materials-14-04541-t001:** Mechanical properties of Hardox 600 steel for 12 mm thick sheet: D—delivery condition, N—normalized condition, L—longitudinal direction, T—transverse direction, PD—producer and distributor data for sheet 6.0–65.0 mm [2,5], OR—the author’s own research results and reported in [4], ND—no data, E—Young’s modulus, R_p0.05_—elastic limit, R_p0.2_—yield strength, R_m_—ultimate tensile strength, A_5_—percentage elongation after fracture for proportional specimens with the original gauge length L_0_ equal to 5 times the cross-sectional area, Z—reduction of area, KCV_+20_—Charpy V-notch toughness at +20 °C, KCV_−40_—Charpy V-notch toughness at −40 °C, HBW—Brinell hardness.

	E	R_p0.05_	R_p0.2_	R_m_	A_5_	Z	KCV_+20_	KCV_−40_	HBW
	[GPa]	[MPa]			[%]		[J/cm^2^]		
D–L (PD)	ND	ND	Min. 1650	Min. 2000	Min. 7	ND	ND	Min. 25	570–640
D–L (OR)	193.9	1302	1544	2106	12.1	38.8	40.3	28.1	559
D–T (OR)	197.4	1324	1572	2123	10.5	37.7	37.8	26.8	553
N–L (OR)	207.1	464	596	848	16.3	46.3	20.1	7.7	317
N–T (OR)	199.2	475	630	907	13.0	44.1	17.8	9.0	264

**Table 2 materials-14-04541-t002:** Chemical composition and carbon equivalents of Hardox 600 steel: PD—producer’s data [5], OR—the author’s own research results and reported in [4], #—sheet thickness, CEV—carbon equivalent according to IIW, CET—carbon equivalent according to SS-EN 1011-2, ND—no data.

	C	Mn	Cr	Ni	Mo	V	Cu	CEV	CET	#
	Selected Chemical Element [% by Weight]							[%]		[mm]
PD	Max. 0.40	Max. 1.00	Max. 1.20	Max. 1.50	Max. 0.60	ND	ND	Max. 0.72	Max. 0.52	3.0–5.0
	Max. 0.47	Max. 1.50	Max. 1.20	Max. 2.50	Max. 0.70			Max. 0.69	Max. 0.57	6.0–35.0
								Max. 0.87	Max. 0.61	35.1–65.0
OR	0.45	0.51	0.33	1.98	0.14	0.009	0.016	0.76	0.58	10.0
	0.44	0.53	0.31	2.03	0.14	0.006	0.011	0.76	0.58	12.0
CEV = C + Mn/6 + (Cr + Mo + V)/5 + (Cu + Ni)/15; CET = C + (Mn + Mo)/10 + (Cr + Cu)/20 + Ni/40										

**Table 3 materials-14-04541-t003:** Chemical composition of Hardox 600 steel: PD—producer’s data [5], OR—the author’s own research results and reported in [4], #—sheet thickness, ND—no data.

	Si	P	S	Al	Ti	Nb	Co	B	#
	Selected Chemical Element [% by Weight]								[mm]
PD	Max. 0.50	Max. 0.015	Max. 0.010	ND	ND	ND	ND	-	3.0–5.0
	Max. 0.70	Max. 0.015	Max. 0.010					0.005	6.0–35.0
									35.1–65.0
OR	0.16	0.012	0.002	0.031	0.006	0.005	0.026	0.0026	10.0
	0.17	0.006	0.002	0.039	0.006	0.000	0.018	0.0022	12.0

**Table 4 materials-14-04541-t004:** Properties of the weld metal used to make the welded joint of Hardox 600 [26]. R_p0.2_—yield strength, R_m_—ultimate tensile strength, A_5_—percentage elongation after fracture for proportional specimens with the original gauge length L0 equal to 4 times the cross-sectional area, KCV_−40_—Charpy V-notch toughness at −40 °C.

Weld	C	Mn	Si	Cr	Ni	Mo	R_p0.2_	R_m_	A_4_	KCV_−40_
	Chemical Composition [% by Weight]						[MPa]		[%]	[J/cm^2^]
OK Autrod 13.43 + OK Flux 10.62	0.11	1.50	0.25	0.60	2.20	0.50	700	800	21	94

**Table 5 materials-14-04541-t005:** Chemical composition of the weld metal obtained by welding over the entire cross-section of the welded joint. X,Y,Z—places marked in Figure 3 where chemical analyzes were performed, CEV—carbon equivalent according to IIW, CET—carbon equivalent according to SS-EN 1011-2.

	X	Y	Z	OK 13.43 + OK Flux 10.62	Hardox 600 Based on Table 2 and Table 3
	Selected Chemical Element [% by Weight]				
C	0.30	0.29	0.30	0.10	0.45
Mn	0.81	0.80	0.80	1.22	0.51
Si	0.23	0.23	0.23	0.30	0.16
P	0.014	0.013	0.014	0.020	0.012
S	0.002	0.002	0.003	0.002	0.002
Cr	0.43	0.43	0.44	0.50	0.33
Ni	2.01	1.99	2.05	1.65	1.98
Mo	0.23	0.22	0.23	0.31	0.14
V	0.009	0.009	0.010	0.009	0.009
Cu	0.040	0.045	0.035	0.105	0.016
Al	0.020	0.019	0.021	0.013	0.031
Ti	0.004	0.004	0.005	0.003	0.006
Nb	0.000	0.000	0.001	0.000	0.005
Co	0.015	0.013	0.018	0.006	0.026
B	0.0020	0.0018	0.0022	0.0014	0.0026
CEV	0.71	0.69	0.71	0.59	0.76
CET	0.48	0.47	0.48	0.33	0.58
CEV = C + Mn/6 + (Cr + Mo + V)/5 + (Cu + Ni)/15; CET = C + (Mn + Mo)/10 + (Cr + Cu)/20 + Ni/40					

**Table 6 materials-14-04541-t006:** Scheme and detailed parameters of thermal treatments as well as selected mechanical properties of welded joints of Hardox 600 steel. E—Young’s modulus, R_p0.05_—elastic limit, R_p0.2_—yield strength, R_m_—ultimate tensile strength, A_5_—percentage elongation after fracture for proportional specimens with the original gauge length L0 equal to 5 times the cross-sectional area, Z—reduction of area, KCV_+20_—Charpy V-notch toughness at +20 °C, KCV_−40_—Charpy V-notch toughness at −40 °C, BM—base material—samples made of Hardox 600 steel sheet thickness 10 mm.

No.	Heat Treatment Scheme and Parameters	E	R_p0.05_	R_p0.2_	Rm	A_5_	Z	KCV_+20_	KCV_−40_
		[GPa]	[MPa]	[MPa]	[MPa]	[%]	[%]	[J/cm^2^]	[J/cm^2^]
1	No treatment	210.3 ± 8.6	565 ± 37	674 ± 43	879 ± 47	6.8 ± 2.6	24.3 ± 13.4	25.0 ± 9.4	6.0 ± 0.1
2	Quenching: 900 °C/20 min./H_2_O Tempering: 100 °C/8 h/Air	207.7 ± 1.7	730 ± 44	1014 ± 28	1046 ± 98	6.5 ± 1.2	20.1 ± 1.3	---	---
3	Normalization: 850 °C/1 h/Air	202.4 ± 6.9	503 ± 18	716 ± 39	885 ± 31	9.5 ± 0.4	44.5 ± 2.8	---	---
4	Normalization: 900 °C/1 h/Air	200.4 ± 2.2	519 ± 32	707 ± 19	873 ± 41	8.1 ± 1.1	43.8 ± 2.4	---	---
5	Normalization: 850 °C/1 h/Air Quenching: 900 °C/20 min./H_2_O Tempering: 100 °C/4 h/Air	208.7 ± 2.7	967 ± 64	1339 ± 94	1810 ± 42	8.9 ± 0.1	40.5 ± 3.7	---	---
6	Normalization: 850 °C/1 h/Air Quenching: 900 °C/20 min./H_2_O Tempering: 150 °C/3 h/Air	192.0 ± 11.7	1183 ± 24	1422 ± 42	1762 ± 42	9.2 ± 0.5	32.0 ± 3.4	50.0 ± 5.5	44.0 ± 5.0
7	Normalization: 850 °C/1 h/Air Quenching: 900 °C/20 min./H_2_O Tempering: 200 °C/3 h/Air	202.0 ± 2.6	1123 ± 23	1424 ± 37	1556 ± 29	9.2 ± 0.9	46.0 ± 0.8	---	---
8	Normalization: 850 °C/1 h/Air Quenching: 900 °C/20 min./H_2_O Tempering: 250 °C/3 h/Air	196.5 ± 1.6	1184 ± 18	1384 ± 27	1478 ± 32	8.8 ± 1.4	45.3 ± 1.0	---	---
9	Normalization: 850 °C/1 h/Air Quenching: 900 °C/20 min./Oil Tempering: 100 °C/20 h/Air	209.7 ± 10.8	1034 ± 28	1354 ± 30	1800 ± 32	7.8 ± 1.3	21.4 ± 9.2	40.0 ± 2.6	32.0 ± 7.8
10	Normalization: 900 °C/1 h/Air Quenching: 900 °C/20 min./H_2_O Tempering: 100 °C/8 h/Air	198.1 ± 6.2	541 ± 24	893 ± 34	1242 ± 101	9.6 ± 1.1	51.3 ± 4.2	---	---
11	Normalization: 900 °C/1 h/Air Quenching: 950 °C/20 min./H_2_O Tempering: 100 °C/4 h/Air	206.1 ± 0.7	1081 ± 79	1518 ± 56	1803 ± 21	8.8 ± 0.3	34.2 ± 5.1	---	---
BM	Normalization: 850 °C/1 h/Air Quenching: 900 °C/20 min./Oil Tempering: 100 °C/20 h/Air	201.0 ± 4.9	999 ± 30	1267 ± 23	2107 ± 17	11.5 ± 0.9	27.5 ± 3.1	33.8 ± 0.9	28.6 ± 9.2

## Data Availability

The data presented in this study are available on request from the corresponding author.

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
