# Peer review of "Technological, Microstructural and Strength Aspects of Welding and Post-Weld Heat Treatment of Martensitic, Wear-Resistant Hardox 600 Steel"

_materials, 2021, doi:10.3390/ma14164541_

Round 1

Reviewer 1 Report

Comment

The paper entitled „Technological, structural and strength aspects of making and heat treatment of welded joints of martensitic, wear-resistant Hardox 600 steel” offers a couple of new information, important for technological treatment (welding and heat treatment) of high wear-resistant martensitic steel.

Questions

Why the author in the Abstracts writesOn the basis of the structural and strength characteristics, the optimal welding technology for Hardox 600 steel was proposed,…”?

In my opinion word “optimal” in the sentence was not in the article demonstrated. There are another technologies of welding, which could be used (laser welding, electron beam welding, combined processes of preheating and welding,…).

Why author writes:

Line 478, Fig. 14 “non-equilibrium ferrite. “ What is it?

Line 698: “…tempering martensite structure with areas of quenching martensite.”?

Fig. 34, 35 and 36 “quenching martensite

Line 707 “…banded quenching martensite - un-tempered martensite.”?

In the Table 6 we can see, that analysed weld was after quenching tempered.

In my opinion terms “quenching martensite” and “un-tempered martensite” are not appropriate.

Suggestions

Line 179 – Location of the blue colour notch in black weld is not visible. White or yellow colour of the notch would be better.

Line 232 and so on – „3 % HNO3 solution“ – probably „3 % Nital“ or „3 % HNO3 in ethyl alcohol“ would be better in whole article

Line 411 – „most optimal“ – „optimal“ without „most“ is appropriate

Line 429 – „steel condition.“ – „as delivered.“ would be better

Line 449 – „perlite“ – „pearlite“ would be better

Line 491 – „FL – fusion line“ is missing in the text or in the line 488

Line 1041 - (Figs. 42b/c and 43b/c). -   probably „(Figs. 42b/d and 43b/d).“ is right.

Author Response

Please, find attached pdf file.

Reviewer 2 Report

The manuscript entitled: Technological, structural, and strength aspects of making and heat treatment of welded joints of martensitic, wear-resistant Hardox 600 steel deals with the Hardox 600 steel weld including its microstructure and properties. The manuscript has been nicely articulated and I recommend the manuscript for publication in the present form.

Author Response

Dear Reviewer,

Thank you very much for conducting a review of the results of my work. I am very glad that the manuscript has gained the approval of the Honorable Reviewer. I tried to present in it as comprehensively and systematically as possible the issue of welding Hardox 600 steel, which in my opinion is still a relatively unknown engineering material, especially in the context of using ultra-high mechanical properties of this steel. Once again, I would like to thank you very much for taking the time to review the research results presented in the manuscript.

Yours sincerely, Łukasz Konat

Reviewer 3 Report

Dear Author, thank you for the opportunity to review the paper on welding processing of wear-resistant Hardox 600 steel.

Due to the ideal combination of high hardness, high strength and excellent toughness, Hardox 600 is suitable for a wide range of applications, such as in forestry, soil cultivation through to road construction. According to the manufacturer, this steel is considered to have good weldability. However, if the welding procedure is not selected correctly, the properties of the welded joint can be negatively affected. In fact, in the literature only general recommendations on the welding procedure can be found. The novelty of the present study is to identify the ways to achieve reliable welded joints of Hardox 600 by applying different regimes of post-weld heat treatment (PWHT).

The paper is well structured. The introduction as well as the cited literature provide a sufficient overview of the problem and state of the art. Nonetheless, I have some comments and suggestions for improvement on the manuscript:

  • Tables 1, 2 and 3 in the introduction part contain material specifications given by the steel manufacturer as well as our own analysis for 12 mm thick sheets, which has already been published in [4]. Please consider whether it is necessary to include this data in the manuscript;
  • it is not clear which base metal was used for the welding tests. The chemical composition given in Tables 1, 2 and 3 differs from the results of the chemical analysis given in Table 5;
  • the same comment concerns the welding consumables. The data in Table 4 for OK 13.43 + OK Flux 10.62 do not match the measurement results in Table 5;
  • another question to the table 5 is where is the measuring area OK 13.43 + OK Flux 10.62 located? The measuring points X, Y, Z are defined in Fig. 3. But it is not clear what OK 13.43 + OK Flux 10.62 means. Is this the whole weld metal? In this case, the measuring points X, Y and Z also belong to it. Please explain these results in more detail;
  • the measurement data in Table 5 is considerable but is not adequately discussed in the text. Some noticeable differences between the individual measured values in zones X, Y, and Z are obvious. Is there a correlation with the welding regions W1, W2, or W3? What is the degree of mixing with the base metal? Please summarize and discuss the results of Table 5;
  • Submerged arc welding (SAW) was used for the welding tests. The applied welding parameters for the welds W1 and W2 result in a heat input of between approx. 1.6 kJ/mm to 2.0 kJ/mm. These values seem to be a little too high, because in welding practice with the Hardox, the welding specification recommends approx. 1 kJ/mm for the 10 mm thick steel plate. Excessive heat input can have a negative effect on the properties of the welded joint. Please explain the selection of the applied welding parameters;
  • The parameter A5 - percentage elongation for the welds was determined during the tests. Was this parameter determined for the pure weld metal? As I know, the values for A5, Rp02 can be determined for homogeneous materials like the base metal or the pure weld metal. Tensile specimens used in this work includes the weld metal with HAZ and the base metal. These areas have different properties. The question is how the values A5, Rp02 were determined for the welded specimens?
  • Please check the writing style und avoid the repetitions like:
    • identify identification, s. Line 131;
    • structural structure, s. Line 706;
    • treatment treatments, s. Line 1110;
    • repetition of "worth mentioning" in two sentences one after another, s. Lines 136, 138

Author Response

Please, find attached pdf file.

Reviewer 4 Report

Dear Authors,

The paper entitle "Technological, structural and strength aspects of making and heat treatment of welded joints of martensitic, wear-resistant Hardox 600 steel" presents valuable information in the field of welding process of the steels.

Materials, methods, results are clearly presented and also the conclusions.

I have some minor recommendations, namely: at fig. 2, 4 the scale is better to be in mm, not in cm; heading 4 Summary is better to be Conclusions

Best regards!

Author Response

Dear Reviewer,

Thank you very much for conducting a review of the results of my work. I would like to thank you very much for taking the time to review the research results in the manuscript. I definitely agree with the suggestions of the Honorable Reviewer. Therefore, all recommended corrections have been done into the manuscript. Again thank you very much.

Yours sincerely, Łukasz Konat

Round 2

Reviewer 3 Report

Dear Author, Thank you for considering the proposed comments.